# Graph is a Substrate Across Data Modalities

**Ziming Li**[1]  **Xiaoming Wu**[2]  **Zehong Wang**[3]  **Jiazheng Li**[1]  **Yijun Tian**[3]
**Jinhe Bi**[4]  **Yunpu Ma**[4]  **Yanfang Ye**[3]  **Chuxu Zhang**[1]

## Abstract

Graphs provide a natural representation of relational structure that arises across diverse domains. Despite this ubiquity, graph structure is typically learned in a modality- and task-isolated manner, where graph representations are constructed within individual task contexts and discarded thereafter. As a result, structural regularities across modalities and tasks are repeatedly reconstructed rather than accumulated at the level of intermediate graph representations. This motivates a representation-learning question: *how should graph structure be organized so that it can persist and accumulate across heterogeneous modalities and tasks?* We adopt a representation-centric perspective in which graph structure is treated as a structural substrate that persists across learning contexts. To instantiate this perspective, we propose **G-Substrate**, a **g**raph **substrate** framework that organizes learning around shared graph structures. G-Substrate comprises two complementary mechanisms: a unified structural schema that ensures compatibility among graph representations across heterogeneous modalities and tasks, and an interleaved role-based training strategy that exposes the same graph structure to multiple functional roles during learning. Experiments across multiple domains, modalities, and tasks show that G-Substrate outperforms task-isolated and naive multi-task learning methods. The codebase, model, and datasets are available at https://github.com/zmli6/G-Substrate.

## 1. Introduction

Graphs provide a natural abstraction for relational information and arise across a wide range of domains and learning

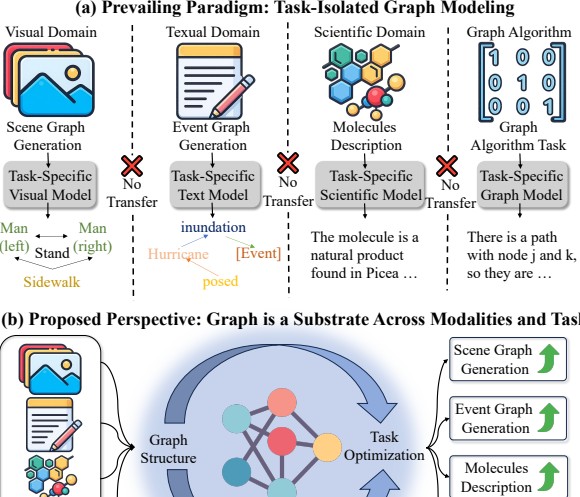

*Figure 1.* **Task-isolated graph modeling vs. graph structure as a substrate.** (a) Graph structure is learned in task-isolated pipelines, causing structurally similar graph patterns to occupy separate representation regions and limit cross-modal interaction. (b) We organize graph structure as a shared substrate, encouraging graph patterns from different data modalities to converge and align, so structurally analogous configurations can mutually shape the representation and improve performance.

problems. For example, in computer vision, scene graphs encode objects and their interactions (Chen et al., 2024d; Li et al., 2024a); in natural language processing, event graphs organize temporal and causal relations (Hu et al., 2025e; Xu et al., 2025b); in chemistry, molecular graphs represent atoms and bonds (Kim et al., 2025; Liu et al., 2024c); and in graph algorithmic tasks, graphs underlie tasks such as connectivity, shortest paths, and structural reasoning (Wang et al., 2025b; Hu et al., 2025c). Despite differences in input modalities and task objectives, graph structure provides an explicit structural interface through which heterogeneous data modalities can be organized (Wang et al., 2025g).

However, the widespread presence of graph-structured representations across tasks and modalities does not imply that learning systems are organized to preserve or accumulate graph structure. In practice, graph structure is built to serve a single objective and discarded after training, as shown in

[1]University of Connecticut [2]National University of Singapore [3]University of Notre Dame [4]LMU Munich. Correspondence to: Chuxu Zhang <chuxu.zhang@uconn.edu>.

*Proceedings of the 43rd International Conference on Machine Learning*, Seoul, South Korea. PMLR 306, 2026. Copyright 2026 by the author(s).

Figure 1(a). Many existing approaches instantiate structure as task-specific graph representations, such as supervision targets in scene graph generation (Wu et al., 2025a; Liu et al., 2025) or event relation extraction (Tao et al., 2025; Zhao et al., 2025c), despite structurally similar patterns recurring across these input modalities. Recent efforts that aim to unify graph-centric learning largely focus on expanding task coverage or sharing model architectures, but still treat graph structure as a task-bound input or output rather than as a persistent intermediate state (Sun et al., 2025; Wang et al., 2024e;b; He et al., 2024). These approaches are architecture-centric: they share model components, but do not establish a graph-level representation state that persists across tasks (Standley et al., 2020; Ruder, 2017). As a result, relational regularities found in one setting do not accumulate at the level of the graph, but remain confined to task-specific formulations. This exposes a representation-level mismatch: graph structure is treated as task-specific data rather than as a persistent learning state.

The above issue motivates the following question: *how should graph structure be organized so that it can persist and accumulate across heterogeneous learning contexts rather than being reconstructed independently in each task?* Moreover, we do not attempt to unify task semantics but rather to align structural patterns that recur across domains.

There are two fundamental dimensions of heterogeneity that prevent graph structure from functioning as a reusable intermediate state, and each motivates a corresponding design requirement. **Heterogeneity in form.** Graph structure varies widely across modalities and tasks in schema, granularity, and representational format (e.g., atom–bond triples in molecules vs. object–relation triples in scene graphs) (Xu et al., 2025a; Chai et al., 2025). This heterogeneity prevents direct reuse of graphs across learning contexts and motivates a *structural compatibility* requirement: graphs from different contexts must be expressible in a common form so that they can coexist in a shared representation space. **Heterogeneity in function.** Graph representations participate in learning under different functional roles: some tasks construct or refine graph structure, while others consume it for reasoning, prediction, or evaluation. A graph optimized under only one role becomes over-specialized to that role. This motivates a *cross-role reuse* requirement: a reusable intermediate graph must remain functional under both structure-generate (Lafferty et al., 2001; Zellers et al., 2018) and structure-understand roles (Battaglia et al., 2018; Hamilton et al., 2017), so that representations are not overfitted to a single objective. Together, these two requirements directly motivate the two complementary mechanisms of G-Substrate: a unified structural schema (addressing form heterogeneity) and interleaved role-based training (addressing function heterogeneity).

Therefore, in this paper, we introduce a representation-centric view which considers *graph structure as a persistent intermediate substrate* for coordinating learning across data modalities and functional roles, as shown in Figure. 1(b). To operationalize this perspective, we introduce G-Substrate, a framework built around two complementary mechanisms: a *unified structural schema* that establishes compatibility of graph representations across tasks and modalities, and *interleaved role-based training* that exposes the same graph to multiple functional roles during learning. These mechanisms address structural and role heterogeneity, respectively.

We evaluate G-Substrate across tasks from multiple domains and modalities and show that it consistently outperforms task-isolated training and naive multi-task baselines. Notably, the unified schema and role-based interleaving play complementary roles: the schema yields gains once multiple tasks share the same graph state space, and role-based interleaving further amplifies these gains by exposing the same graph to multiple functional roles. Their combination consistently yields the strongest performance, suggesting that the most robust graph representations emerge when structural alignment is coupled with role-based training.

## 2. The G-Substrate Framework

This section presents the G-Substrate framework and describes how the substrate-oriented perspective is realized in both data representation and model learning. Specifically, we formalize the central perspective of this work: *graph structure as a persistent substrate rather than a task-bound artifact* (Section 2.1). This perspective leads to two design requirements, namely structural compatibility and cross-role reuse, which together define the design space of the framework. We address the first requirement by organizing graphs within a unified graph state space, aligning representations from heterogeneous tasks into a common structural form (Section 2.2). We address the second requirement through interleaved role-based supervision, a training organization that exposes graph to multiple functional roles and promotes their reuse across learning contexts (Section 2.3).

### 2.1. Perspective: Graph is a Structural Substrate

Graph structure arises across a wide range of learning problems, but is most often modeled in a task-bound manner. In many settings, such a structure is made explicit through graph representations. In prevailing practice, graph representations are constructed to serve individual task objectives and discarded thereafter, causing structural regularities that recur across tasks and modalities to be repeatedly reconstructed in isolation. In contrast, we introduce a new representation-centric perspective: **graph is a reusable structural substrate across data modalities.** Building on this perspective, we propose G-Substrate, a framework that organizes learning contexts across domains and modalities.

*Table 1.* Coarse topology statistics (per-graph averages). While global structural scale differs, recurring local structures are observed across all domains.

| Domain | AvgDeg | ASPL | TwoHop | Hubs |
|---|---|---|---|---|
| Graph algorithm | 5.8 | 2.1 | 633 | 16 |
| Molecular graphs | 2.1 | 6.2 | 53 | 12 |
| Scene graphs | 1.5 | 1.4 | 2.4 | 0.7 |
| Event graphs | 1.5 | 1.4 | 15 | 0.9 |

To empirically support this perspective, we examine whether structurally similar graph configurations recur across heterogeneous tasks and whether they play comparable structural roles despite differences in task semantics. We provide evidence for this perspective through quantitative structural statistics and qualitative motif analysis across heterogeneous domains, as reported in Table 1. These statistics summarize coarse topological properties, including average degree (AvgDeg), average shortest path length (ASPL), and the prevalence of simple local motifs such as two-hop chains and hub-centered patterns. Although global graph properties differ substantially, coarse local structures recur with non-trivial frequency in all settings studied. Beyond their prevalence, these graph structures play aligned functional roles. Figure 2 shows a hub-centered configuration in an event graph and a scene graph. In the former, the event *received* participates in multiple temporal dependencies; in the latter, the object *horse* participates in multiple spatial relations. While task semantics differ, the central node in both cases coordinates multiple edges and constrains how they compose, indicating cross-domain invariance at the level of graph structure rather than task-specific meaning. These observations are drawn from representative datasets in scene graph generation, event relation extraction, molecular graphs, and algorithmic graph tasks, with detailed dataset descriptions and measurements provided in Appendix A.

To formalize this substrate-oriented view, we treat a graph as the fundamental structural representation. Specifically, a graph is defined as a set of structural triples $G = \{(u, r, v)\}$, where $u$ and $v$ denote entities and $r$ denotes a typed edge between them. The relation label $r$ and entity identities are preserved as part of the structural representation; relations such as *before* or *wearing* retain their inherent meaning. What this definition excludes is *task-specific framing*, such as loss functions, execution logic, and optimization objectives, rather than the semantic content of relations and entities themselves. A graph's identity is determined solely by the structural configuration it encodes, not by how a particular task consumes it. Entities or edges may carry optional attributes, which serve as auxiliary annotations and leave the relational structure unchanged.

The graph substrate perspective treats graph as an intermediate structural representation intended to persist across learning contexts. For a graph to serve this role, two require-

ments follow. First, graphs arising from different learning settings must be structurally compatible so that they can reside in a unified representation space. Second, training must explicitly support the reuse of graphs across functional roles, rather than confining them to task-local roles. G-Substrate operationalizes these requirements through a unified structural schema and an interleaved cross-task training strategy.

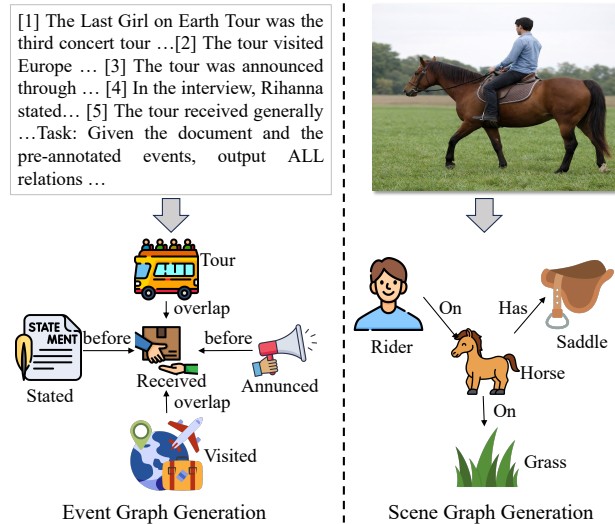

Event Graph Generation Scene Graph Generation

*Figure 2.* Analogous constraint roles of hub motifs across tasks. In the event graph, the hub event *received* participates in multiple temporal dependencies; in the scene graph, the hub object *horse* participates in multiple spatial relations. The central node coordinates multiple relations and constrains their joint consistency.

## 2.2. Structural Compatibility: A Unified Schema

To ensure structural compatibility across tasks, G-Substrate organizes graphs in a *unified graph state space*. Building on the graph definition in Section 2.1, we denote this space as $\mathcal{G}_s = \{ G \mid G = \{(u, r, v)\} \}$, where each $G \in \mathcal{G}_s$ consists of entities $u, v$ connected by typed edges $r$. Importantly, $\mathcal{G}_s$ is not the universal set of all conceivable graphs, but a *structured family* constrained by the unified schema: all elements share consistent node identifiers, typed edges following fixed conventions, and the same $(u, r, v)$ triplet format. Graphs that do not conform to these conventions lie outside $\mathcal{G}_s$. This constraint is what gives the space its utility as a shared representation: only by restricting $\mathcal{G}_s$ to a structured family do graphs from heterogeneous tasks become directly comparable and reusable. Graphs arising from different modalities and tasks are mapped into this common structural space, sharing consistent node identifiers, edge types, and connectivity rules. Figure 2 gives examples of this mapping. An event graph constructed from text and a scene graph constructed from an image are both represented as graph instances $G \in \mathcal{G}_s$. Although originating from different modalities and tasks, these graphs share the same structural form, making hub-centered structural

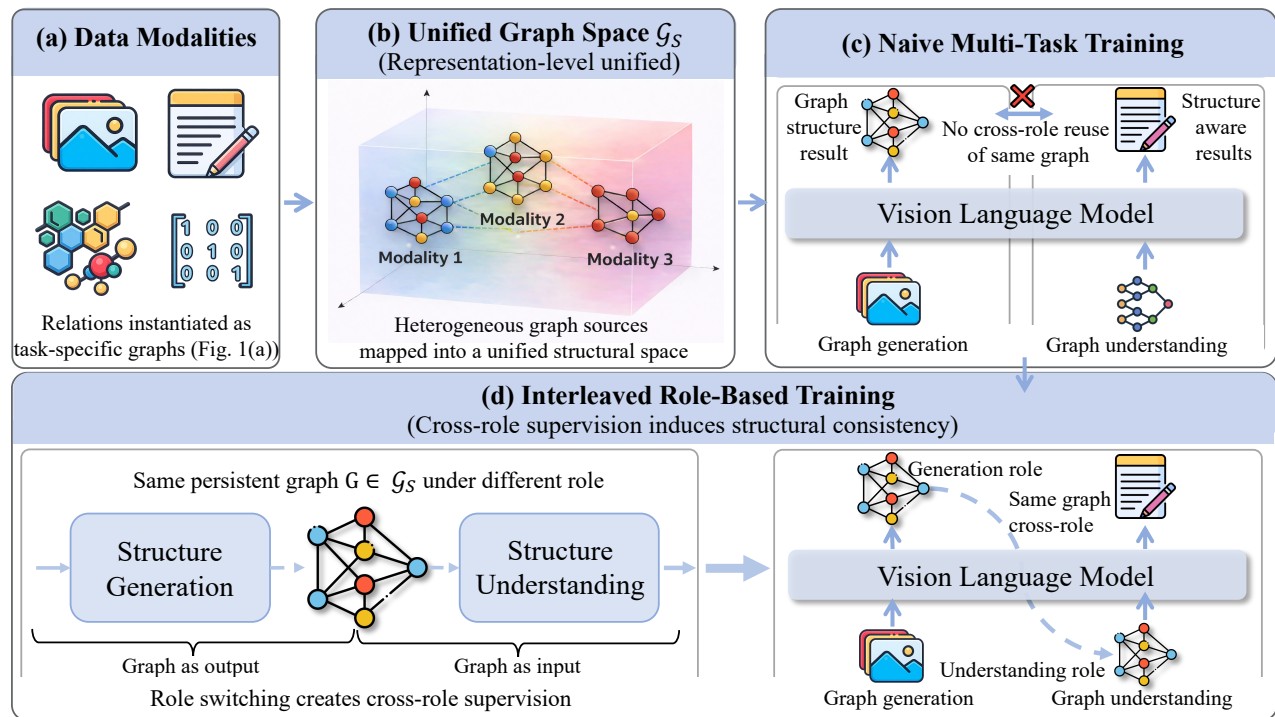

*Figure 3.* **Unified graph substrate and cross-role training.** Graph structures from heterogeneous modalities are mapped into a unified graph state space $\mathcal{G}_s$, where graphs serve as persistent structural representations (b). Under naive multi-task training (c), graphs remain confined to fixed task roles, and the same graph is not reused across functional contexts. Our interleaved role-based paradigm (d) exposes the *same* graph $g \in \mathcal{G}_s$ to both structure-generation and structure-understanding roles, creating cross-role supervision. This role switching induces structural consistency and supports reusable graph representations across tasks and modalities.

patterns (e.g., *received* and *horse*) comparable in $\mathcal{G}_s$.

Figure 3(b) provides a geometric intuition for this alignment. Graphs from different modalities and tasks may initially occupy disjoint regions under task-specific conventions. Expressing them in the unified graph state space $\mathcal{G}_s$ brings these heterogeneous constructions into a common structural region, where structurally compatible patterns become aligned. Importantly, this concentration arises from explicit structural representation alignment rather than from parameter sharing or feature similarity alone.

Structural compatibility alone, however, does not guarantee that graphs are meaningfully exercised under diverse functional roles during learning. Even in a shared structural space, a graph may still be optimized only in a single usage context. Enabling unified graph representations to function consistently across roles and tasks therefore requires an appropriate training organization, which is provided by the interleaved role-based training described next.

### 2.3. Cross-task Reuse: Interleaved Role-based Training

A unified graph state space establishes structural compatibility of graphs across tasks, but does not by itself determine how those graphs are used during learning. Under naive multi-task training, different tasks are optimized jointly

using a shared backbone model, with each task receiving its native modality input and producing task-specific outputs under its own objective. A common instantiation of this naive setup is to use a single vision–language foundation model as a shared backbone to process heterogeneous modalities, with lightweight task-specific heads applied for different tasks (Wei et al., 2024; Zhu et al., 2025). Although graphs may implicitly arise in the shared model, they function primarily as task-internal intermediates rather than as persistent representations reused across tasks. As illustrated in Figure 3(c), graphs generated or used in one task do not typically participate in other task roles or learning contexts. Consequently, even when tasks operate in a common graph state space, their usage of graphs remains largely isolated, resulting in limited cross-role interaction.

To enable graphs to function as persistent intermediate representations across functional contexts, training must explicitly organize how they are reused under different task types. Interleaved generation–understanding training pipeline provides this organization. We model training as a sequence of *task–role instantiations* over unified graph states. Let $\mathcal{T} = \{T_1, \ldots, T_K\}$ denote a set of tasks under different modalities, and $\mathcal{G}_s$ denotes the unified graph state space.

Each task $T_i$ is associated with a role function

$$\rho_i : \mathcal{G}_s \to \{\text{GENERATE}, \text{UNDERSTAND}\}, \qquad (1)$$

where GENERATE corresponds to tasks that construct or refine the graph structure (e.g., scene graph generation, event graph extraction), and UNDERSTAND corresponds to tasks that operate on graph structure for reasoning, prediction, or evaluation (e.g., graph algorithm).

Next, training is organized as a sequence $\{(T_{i_t}, \rho_{i_t})\}_{t=1}^{N}$, in which graphs produced under generation tasks may be reused as inputs under subsequent understanding tasks. To make the input–output flow explicit, we view each task $T_i$ as an operator acting on graphs and, optionally, modality inputs. Let $\mathcal{X}_i$ denote the modality-specific input space associated with $T_i$, and let $\mathcal{Y}_i$ denote its task-specific output space. Each task induces a mapping

$$T_i : (\mathcal{X}_i, \mathcal{G}_s) \to (\mathcal{G}_s, \mathcal{Y}_i). \qquad (2)$$

When $\rho_i = \text{GENERATE}$, the task produces or refines a graph:

$$G^{(t)} = T_i(x_i, G^{(t-1)}), \quad G^{(t)} \in \mathcal{G}_s. \qquad (3)$$

When $\rho_i = \text{UNDERSTAND}$, the task treats the graph as an intermediate representation and produces predictions or supervision signals:

$$y_i^{(t)} = T_i(x_i, G^{(t)}), \quad y_i^{(t)} \in \mathcal{Y}_i. \qquad (4)$$

Interleaving therefore induces a trajectory of a graph. $G^{(0)} \to G^{(1)} \to \cdots \to G^{(N)}$, where graphs serve as persistent intermediate representations that evolve across successive generations and understanding tasks rather than being reconstructed independently and discarded in each task, as illustrated in Figure 3(d).

**Concrete example.** Consider a two-step trajectory with role sequence (GENERATE, UNDERSTAND). At $t = 0$, $G^{(0)}$ is the empty graph. At $t = 1$, given a street-scene image $x_1$, a scene graph generation task produces $G^{(1)} = \{(rider, on, horse), (horse, on, grass), (rider, wearing, helmet)\}$. At $t = 2$, a graph reasoning task operates directly on $G^{(1)}$: given the query *"is there a path from **rider** to **grass**?"*, the model traverses $G^{(1)}$ along $rider \xrightarrow{on} horse \xrightarrow{on} grass$ and returns $y^{(2)} = yes$. The graph $G^{(1)}$ itself is not modified. The same graph state is thus produced under GENERATE by a vision task and immediately reused under UNDERSTAND by a structural reasoning task.

Schema compatibility extends this reuse beyond a single modality. For instance, extracting typed temporal relations (e.g., *before*, *overlap*) from a news passage yields an event graph whose triplets share the $(u, r, v)$ format with $G^{(1)}$, so both reside in $\mathcal{G}_s$ despite differing modalities and relation vocabularies.

From a representation-centric perspective, interleaving alters the supervision received by graphs rather than modifying individual task objectives. In task-isolated training, a graph is optimized under a single task type and only needs to satisfy constraints induced by that usage context. Under interleaved generation–understanding training, the same graph must remain usable across multiple task types. Graphs that support one type but are structurally incompatible with others receive inconsistent supervision and are gradually disfavored. This bias toward structurally coherent graphs emerges from the training organization itself, rather than from explicit regularization or parameter-level coupling.

## 3. Experiments

This section examines whether organizing learning around reusable intermediate graph leads to consistent improvements across heterogeneous learning settings mediated by graph structure. We study this question by contrasting G-Substrate with task-isolated training and naive multi-task learning, and by conducting controlled analyses that disentangle the roles of structural alignment and cross-role reuse of graph. We additionally assess how a unified, substrate-oriented framework compares to representative task-specific models under standard evaluation protocols.

### 3.1. Learning Settings and Tasks

We evaluate the framework on four representative learning settings spanning domains and modalities. For each task, we describe its objective, model inputs and outputs, datasets, evaluation metrics, and task-specific baselines.

**Graph Algorithmic Reasoning (GAR).** This task predicts the outputs of classical graph algorithms from an input attributed graph. The model takes an attributed graph as input and outputs the answer to a graph algorithmic query. We consider connectivity (CT), cycle detection (CD), shortest path (SP), and bipartite matching (BM). We follow the datasets and evaluation settings in prior work (Wei et al., 2024; Wang et al., 2024b), and report accuracy as the evaluation metric. We compare against representative task-specific models for graph algorithmic reasoning, including GITA (Wei et al., 2024) and GraphWiz (Chen et al., 2024b).

**Molecular Graph Description (MGD).** This task requires generating a natural-language description of a molecule from its structural representation. The model takes a molecular graph (atoms and bonds), optionally accompanied by its SMILES string, as input and outputs a textual description of molecular properties or functionality. We use the Mol-Instructions dataset (Fang et al., 2024), and evaluate using BLEU-4 and ROUGE-L. We compare against the task-specific baseline Mol-LLaMA (Kim et al., 2025).

**Scene Graph Generation (SGG).** This task requires predicting a scene graph of objects and relations from an input image. The model takes an image as input and outputs a

*Table 2.* **Main results across modalities, domains, and tasks.** Best results are in **bold**; second-best are underlined. GAR (Graph Algorithmic Reasoning) is evaluated using **accuracy** for each task (CT: Connectivity, CD: Cycle Detection, SP: Shortest Path, BM: Bipartite Matching). MGD (Molecular Graph Description) is evaluated using **BLEU-4** and **ROUGE-L**. SGG (Scene Graph Generation) reports **PCIs R@50**. ERE (Event Relation Extraction) reports **F1 scores** on MAVEN-S, MAVEN-T, MAVEN-C, and HiEvent.

| | GAR | | | | MGD | | SGG | ERE | | | |
|---|---|---|---|---|---|---|---|---|---|---|---|
| **Method** | CT | CD | SP | BM | BLEU-4 | ROUGE-L | PCIs | MA-S | MA-T | MA-C | HiE |
| **Task-Specific Training** | | | | | | | | | | | |
| GITA (Wei et al., 2024) | 98.17 | **98.07** | 39.15 | 93.19 | – | – | – | – | – | – | – |
| G-Wiz (Chen et al., 2024b) | 97.74 | 95.46 | 41.46 | 92.15 | – | – | – | – | – | – | – |
| M-LLama (Kim et al., 2025) | – | – | – | – | 50.74 | 67.02 | – | – | – | – | – |
| PGSG (Li et al., 2024a) | – | – | – | – | – | – | **26.9** | – | – | – | – |
| ProtoEM (Hu et al., 2023) | – | – | – | – | – | – | – | 53.80 | 31.80 | 27.90 | 20.43 |
| LLMERE (Hu et al., 2025e) | – | – | – | – | – | – | – | **54.30** | 35.60 | 27.90 | 22.90 |
| Naive single-task | 99.44 | 92.18 | 38.27 | 92.05 | 48.59 | 66.65 | 23.74 | 39.65 | 41.60 | 27.70 | 17.10 |
| Unified single-task | 97.80 | 94.70 | 37.14 | 85.98 | 47.35 | 65.64 | 22.43 | 45.45 | 33.29 | 30.22 | 14.28 |
| **Multi-Task Training** | | | | | | | | | | | |
| Naive multi-task | **99.71** | 94.72 | 41.27 | 92.21 | 48.11 | 66.11 | 24.68 | 36.87 | 39.14 | 37.02 | 18.78 |
| Unified multi-task | 98.09 | 96.19 | 45.02 | 94.23 | 49.99 | 67.36 | 25.36 | 51.89 | 40.05 | 40.75 | 19.37 |
| Naive multi-task + interleave | 98.27 | 93.86 | 43.83 | 91.92 | 48.63 | 64.98 | 24.02 | 45.74 | 38.86 | 37.99 | 21.36 |
| **G-Substrate (Ours)** | 98.41 | 96.97 | **48.59** | **94.54** | **51.53** | **68.47** | 25.38 | 52.20 | **42.68** | **40.91** | **25.15** |

structured graph whose nodes correspond to objects and whose edges represent pairwise relations. Evaluation is conducted on Visual Genome (Krishna et al., 2017) under the PCIs and SGCLs protocols, reporting R@50 and mR@50, with PCIs R@50 as the primary metric. As ground-truth bounding boxes are unavailable in our setting, we follow the data processing protocol of (Li et al., 2024a). We compare against the task-specific baseline PGSG (Li et al., 2024a).

**Event Relation Extraction (ERE).** This task constructs event-relation graphs from text, capturing temporal, causal, or subevent structures among events. The model takes raw text as input and outputs a structured graph whose nodes correspond to events and whose edges encode typed relations. We evaluate on MAVEN-subevent (MA-S), MAVEN-temporal (MA-T), MAVEN-causal (MA-C) (Wang et al., 2022) and HiEvent (HiE) (Glavas et al., 2014), reporting precision, recall, and F1 score. We compare against task-specific baselines ProtoEM (Hu et al., 2023) and LLMERE (Hu et al., 2025e).

### 3.2. Training Paradigms

We compare learning settings that differ along two orthogonal axes: (1) the representation of graph (task-specific vs. unified structural schema), and (2) the training organization (task-isolated, jointly multi-task, or with interleaved role-based training). This leads to six paradigms. **Naive single-task (NST)** and **Unified single-task (UST)** train each task in isolation, differing only in whether graph uses native formats or the unified schema. **Naive multi-task (NMT)** and **Unified multi-task (UMT)** jointly train all tasks, again differing in representation format but without exposing the same graph to multiple functional roles. **Naive multi-task +**

**interleave (NMT-I)** introduces role-based interleaving on top of naive task-specific representations, allowing the graph to be reused under different task roles without structural alignment. **G-Substrate (Unified + interleave, G-Sub)** combines the unified schema with interleaved role-based training. Together, these paradigms disentangle the effects of structural alignment and cross-role reuse. Detailed definitions are given in Appendix D. All methods share the same backbone model and optimization settings. For multi-task settings, no additional task-specific fine-tuning after training or test-time adaptation is applied; each model is trained once under its corresponding paradigm and evaluated directly. Unless otherwise specified, experiments use the Qwen3-VL-2B-Instruct model (Team, 2025) as the backbone. Detailed training configurations are provided in Appendix E.

### 3.3. Main Results

Table 2 summarizes the main results. Although G-Substrate uses a single unified model rather than domain-specialized architectures, it matches or exceeds task-specific systems on most metrics. On GAR, SP rises from G-Wiz's 41.46 to 48.59. On MGD, it reaches 51.53 BLEU-4 and 68.47 ROUGE-L, exceeding M-LLaMA's 50.74 and 67.02. On ERE, F1 on MA-T, MA-C, and HiE rises from LLMERE's 35.60, 27.90, and 22.90 to 42.68, 40.91, and 25.15. The gains are largest where evaluation rewards relational reasoning rather than local pattern matching, such as SP, BM, and multi-hop event relations, and smallest in structurally compact settings, where PGSG still leads SGG, 26.9 to 25.38, and LLMERE retains a narrow edge on MA-S, 54.30 to 52.20. We interpret this as evidence that organizing learning around a shared substrate carries enough structural induc-

tive bias to match domain-specialized pipelines without sacrificing per-domain capability, while leaving room for task-specific tuning where graphs are small enough that specialization itself is the dominant lever.

We next analyze the effect of different training paradigms. G-Substrate outperforms both task-isolated training and naive multi-task learning on most metrics. The improvements are more pronounced in settings with stronger structural demands, suggesting that the gains are tied to structural reasoning rather than uniform scaling effects. These patterns are consistent with the intended mechanism of G-Substrate. Task-isolated training restricts graphs to a single functional context, while naive multi-task learning, despite parameter sharing, does not require the same graph to remain usable across roles. By contrast, G-Substrate combines structural alignment with interleaved generation–understanding training, encouraging graphs to remain valid under multiple roles. This cross-role pressure biases representations toward relational regularities rather than task-specific shortcuts, aligning with the observed performance trends. Detailed results are provided in Appendix F. We additionally compare against the gradient-balancing multi-task baseline GradNorm in Appendix I, where G-Substrate outperforms NMT+GradNorm across all four domains without any explicit loss reweighting, indicating that the dominant bottleneck is representational rather than optimization-level.

### 3.4. Analysis

We conduct controlled studies to analyze the mechanisms underlying G-Substrate, isolating representation and training-organization factors while keeping the backbone, data, and training budget fixed. Specifically, we examine: (i) the interaction between structural alignment and role-based training, (ii) the effect of schema realization, (iii) cross-domain structural transfer, (iv) the contribution of different cross-role training instantiations, (v) the role of structural correctness of the reused graph, and (vi) the impact of the proportion of role-based interleaving.

#### 3.4.1. UNIFIED STRATEGY ANALYSIS

**Schema–Training Interaction.** We examine whether the effect of unified representations arises from the structural schema itself or from its interaction with role-based training. Table 2 shows that the *Unified Single-Task* setting does not outperform the *Naive Single-Task* baseline and often performs worse under task-isolated training. By contrast, in the multi-task setting the unified schema yields consistent improvements over its naive counterpart (*Unified Multi-Task* vs. *Naive Multi-Task*), and these gains are further amplified once the same graph is exposed to multiple functional roles during training. This indicates that the schema primarily establishes structural compatibility, whose benefits emerge

once graphs are shared across tasks and grow strongest under role-based reuse.

**Effect of Schema Realization.** We compare different realizations of the unified schema, including natural-language descriptions, XML-style serializations, and the schema representation used in G-Substrate, all encoding an identical graph under the same role-based training setting. Table 3 shows that although alternative serializations permit basic transfer, their performance is generally less stable. XML-style formats, in particular, tend to underperform, likely because strict formatting encourages attention to surface structure rather than underlying relational semantics. The proposed schema realization provides more reliable performance, indicating that effective structural reuse depends not only on schema unification, but also on how relational structure is expressed when the graph is exercised under multiple functional roles during training.

**Cross-domain Structural Transfer.** To assess cross-domain reuse of graph structure, we transfer from event-centric text graphs to scene graph generation. Table 4 reports performance relative to a base model without domain-specific pretraining. Training on event graphs alone improves scene graph generation despite the absence of target-domain supervision. This suggests that learning organized around an explicit graph structure can capture structural regularities that transfer across domains, rather than being fully tied to a single task or modality.

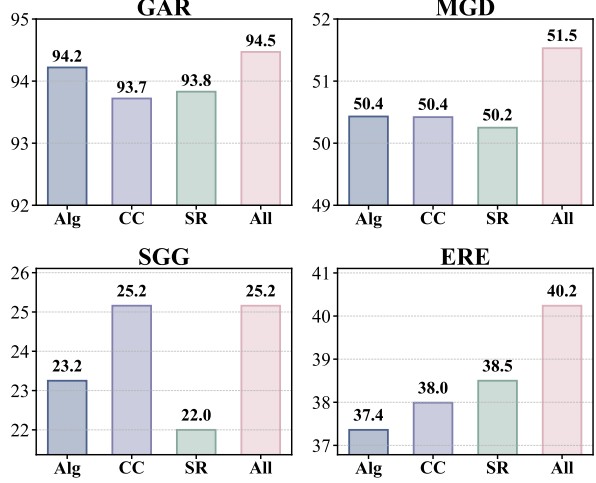

*Figure 4.* **Contribution of different interleaving supervision types.** Metrics are averaged accuracy for GAR, BLEU-4 for MGD, PCIs R@50 for SGG, and macro-averaged F1 for ERE.

#### 3.4.2. INTERLEAVING STRATEGY ANALYSIS

**Cross-task Influence.** To analyze how different cross-role training instantiations contribute to learning, we vary the composition of role-based exposure sources while keeping the overall training budget fixed. Specifically, we consider three types of role-based supervision: graph algorithmic

*Table 3.* **Effect of Schema Realization.** Performance comparison of different schema realizations under identical multi-task training conditions. The best-performing method is shown in **bold**.

| Method | GAR | | | | MGD | | SGG | ERE | | | |
|---|---|---|---|---|---|---|---|---|---|---|---|
| | CT | CD | SP | BM | BLEU-4 | ROUGE-L | PCIs | MA-S | MA-T | MA-C | HiE |
| Natural Language | 97.15 | 94.20 | 44.80 | 92.75 | 49.20 | 66.80 | 24.10 | 50.80 | 40.95 | 39.20 | 23.90 |
| XML-style | 94.80 | 92.30 | 40.10 | 88.40 | 44.60 | 60.50 | 23.65 | 46.30 | 36.40 | 34.70 | 20.80 |
| **Ours** | **98.41** | **96.97** | **48.59** | **94.54** | **51.53** | **68.47** | **25.38** | **52.20** | **42.68** | **40.91** | **25.15** |

*Table 4.* **Cross-domain structural transfer from event graphs to scene graph generation.** Models are evaluated on scene graph generation (PCIs R@50). $\Delta$ denotes the absolute performance change relative to the Base model. No scene-graph data is used during source-domain training.

| Pretraining Setting | SGG | $\Delta$ |
|---|---|---|
| Base (no domain training) | 19.10 | – |
| Event-only (unified schema) | 21.47 | +2.37 |

(Alg), consistency checking (CC), and subgraph retrieval (SR), together with their combination. All three instantiate the UNDERSTAND side of the role function $\rho$ defined in Section 2.3: each takes an existing graph $G \in \mathcal{G}_s$ as input and produces a non-graph prediction, thereby complementing the GENERATE side already covered by SGG and ERE among the main tasks and exposing the same persistent graph to multiple functional roles. Alg requires structural reasoning over graphs (e.g., connectivity), encouraging preservation of global structure. CC presents the original modality input (text or image) together with a candidate graph and predicts whether they are consistent; negative examples are constructed by perturbing the graph, promoting alignment between graph structure and underlying inputs. SR operates on scene and event graphs, requiring the model to recognize structurally meaningful subgraphs, encouraging localized structural reasoning and compositional reuse. Figure 4 shows the resulting performance changes relative to unified multi-task training without role-based interleaving. Gains are not uniform, but relate systematically to the domain structural characteristics: highly constrained graph domains show smaller improvements, whereas more weakly constrained domains benefit more from additional cross-role structural exposure. Effects also depend on the supervision type, with consistency checking and subgraph retrieval often yield stronger gains, particularly when supervision is grounded in the same evidence modality. These trends indicate that role-based interleaving reshapes representation-level structural pressures on the persistent graph rather than uniformly enhancing all tasks.

**Structural Correctness of Reused Graphs.** We test whether the gains from role-based interleaving depend on structural coherence rather than superficial serialization. Persistent graphs reused under multiple functional roles are replaced with structurally incorrect variants that preserve

node and edge labels but disrupt relational connectivity. As shown in Figure 5, performance gains largely disappear when structurally incorrect graphs are used. This contrast indicates that cross-role training is sensitive to the relational organization of the graph, and that malformed structures introduce misleading signals at the representation level.

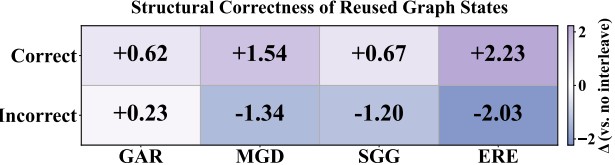

*Figure 5.* **Effect of structural correctness of reused graph.** Performance change ($\Delta$ vs. unified multi-task) for structurally correct and incorrect graphs. Metrics are averaged accuracy for GAR, BLEU-4 for MGD, PCIs R@50 for SGG, and macro-averaged F1 for ERE.

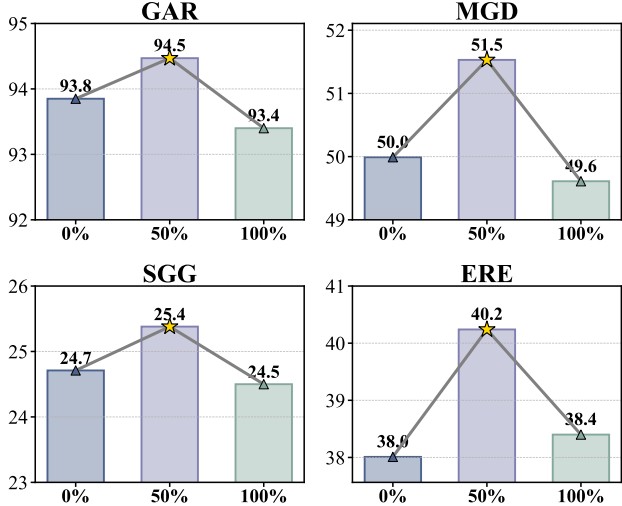

*Figure 6.* Effect of interleaving proportion. Performance of each domain as the ratio of newly introduced interleaved training instances to the unified multi-task data increases (0, 50%, 100%). Metrics are averaged accuracy for GAR, BLEU-4 for MGD, PCIs R@50 for SGG, and averaged F1 for ERE.

**Effect of Role-based Interleaving Proportion.** Finally, we analyze how the relative proportion of role-based exposure affects performance. As shown in Figure 6, we vary the ratio of training instances in which persistent graphs are exercised under multiple functional roles to those drawn from standard unified multi-task training. Moderate levels of cross-role exposure consistently yield the greatest improvements, whereas excessive role-based interleaving

degrades performance by weakening task-specific optimization signals. This trend indicates that effective role-based training requires balancing structural exposure of the graph across roles with sufficient task-focused learning. Together, these results suggest that role-based interleaving operates as a controlled mechanism for representation reuse rather than as unrestricted task mixing. Results with an alternative backbone show similar trends (Appendix G), suggesting the gains arise from representation design and training organization rather than the backbone.

### 3.4.3. ROBUSTNESS TO NOISY GRAPH EXTRACTION

In realistic pipelines, scene graph generators, event extractors, and parsers all produce structurally imperfect graphs. To probe whether G-Substrate's gains depend on clean extractions, we perturb a fixed proportion of edges in the graphs reused during interleaved training while leaving primary task data unperturbed. The perturbations include relation-label replacement, entity substitution, subject–object swapping, and edge deletion. Full results across noise levels $\{0\%, 10\%, 20\%, 30\%\}$ are in Appendix J. We observe graceful degradation. Under 20% corruption, G-Substrate still exceeds clean NMT by 50.74 to 48.11 on MGD and by 39.74 to 38.02 on ERE, and it remains competitive on GAR; even at 30% noise, three of four domains stay close to or above clean NMT. SGG is the exception, being more sensitive at all noise levels, likely because scene graphs are structurally compact. Consistent with Figure 5, complete corruption reverses gains but partial noise does not, indicating that cross-role reuse acts as a structural regularizer: only patterns reinforced across roles are retained, so noise in any single context cannot dominate the learned representation.

## 4. Related Work

**Graphs as a ubiquitous but task-bound tool.** Graph-structured representations have become a standard modeling device across diverse domains (Wang et al., 2025b; Hu et al., 2025c; Chen et al., 2024b; Wang et al., 2024b; 2023; Yuan et al., 2025b; Zhang et al., 2019b; Ju et al., 2022). Some works focus on inducing graphs from perceptual or linguistic inputs, such as scene graph generation from images and event graph extraction from text (Chen et al., 2024d; Li et al., 2024a; Liu et al., 2025; Xu et al., 2025c; Zhang et al., 2022). Other works adopt graph-conditioned reasoning paradigms, including molecular understanding and structured semantic prediction (Kim et al., 2025; Liu et al., 2024c; Park et al., 2024; Guo et al., 2021; 2020). Despite the recurrence of similar structural patterns across domains, existing systems almost universally treat graph structure as a *task-scoped artifact*: graphs are constructed to satisfy a particular objective, optimized within a single

task pipeline, and discarded thereafter. Consequently, graph representations do not function as reusable state across heterogeneous learning contexts, and structural regularities are repeatedly rediscovered rather than accumulated.

**Multi-task learning.** Multi-task learning (MTL) has long been studied as a paradigm for coordinating learning across related tasks (Ruder, 2017; Akhtar et al., 2020; Yuan et al., 2025a; Zhang & Yang, 2022; Sanh et al., 2021). More recently, large language models and vision–language models have significantly extended this paradigm by leveraging large-scale pretraining, unified architectures, and instruction-based or prompt-based task formulations to support broad task generalization (Kong et al., 2025; Sun et al., 2025; Wang et al., 2024e; He et al., 2024; Liu et al., 2024b; Wang et al., 2026; Lu et al., 2019; Tan & Bansal, 2019; Chen et al., 2020). Despite these advances, most existing LLM- and VLM-based multi-task frameworks rely on implicit knowledge storage in model parameters (Zhang et al., 2026; Khashabi et al., 2020; Mishra et al., 2022), enabled by shared objectives and architectures, rather than on explicitly modeling and reusing structured representations across tasks. Consequently, while effective at multi-task prediction, they fall short of exploiting recurring graph structures across tasks as an explicit and reusable source of inductive bias.

**Graph as a unified substrate across modalities.** We argue that existing limitations stem from representation design. We therefore treat graph structure as a persistent intermediate state shared across domains and modalities, instead of a task-bound interface, enabling structural knowledge to accumulate and transfer across learning tasks. Appendix H provides further discussion.

## 5. Conclusion

Graph structures arise across diverse domains, modalities, and tasks, but are typically optimized in isolated learning contexts and discarded thereafter, preventing them from serving as persistent intermediate representations. We argue that this limitation stems from a task-centric organization of learning that treats intermediate structure as disposable rather than reusable. To address this issue, we introduce G-Substrate, a framework that enables representation reuse through two complementary mechanisms: a unified structural space that ensures cross-task compatibility, and interleaved role-based training that exposes the same graph to multiple functional roles. Experiments across heterogeneous settings show that the unified structural space yields gains once multiple tasks share the representation, role-based interleaving further amplifies these gains, and their combination yields the most consistent improvements. Together, these findings indicate that persistent graph representations are a key driver of structural reuse and improved performance across diverse learning contexts.

# Acknowledgments

This work was partially supported by the NSF under grants IIS-2528540, IIS-2334193, IIS-2340346, CNS-2426514, and CMMI-2146076. This work also used computational resources provided through NSF ACCESS grant CIS260048. Any opinions, findings, conclusions, or recommendations expressed in this material are those of the authors and do not necessarily reflect the views of the sponsors.

# Impact Statement

We propose a representation-centric framework that treats graph structure as a reusable intermediate substrate across tasks and modalities, with potential benefits for data efficiency and generalization in AI systems operating over structured information such as events, scenes, molecules, and algorithmic graphs.

A potential negative impact is that, because graph states are reused across tasks, systematic biases in how entities or relations are represented (e.g., in event corpora or visual datasets) may propagate across domains rather than remaining task-local. Responsible deployment therefore depends on careful dataset composition, transparent graph construction, and cross-domain evaluation. The present work also does not explicitly study how the composition of modalities, domains, or role types shapes representation formation; future work may explore principled strategies for balancing heterogeneous role-based supervision and extending this representation-centric principle to other forms of structured intermediate representations beyond graphs.

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

# Appendix

## A. Empirical Motivation: Recurrence of Structural Motifs Across Domains

This appendix provides empirical motivation for the representation-centric perspective in Section 2.1. We analyze whether coarse structural motifs recur across heterogeneous domains when relational structure is expressed as graph states, i.e., sets of relational tuples $\mathcal{G} = \{(u, r, v)\}$. Our goal is to verify that (i) simple local motifs (e.g., two-hop chains and hub structures) appear consistently across domains, while (ii) global structural scales (e.g., path lengths) can vary in a domain-dependent but interpretable manner.

### A.1. Datasets and Graph View

We analyze four domains used throughout the paper: graph algorithm (Wei et al., 2024; Wang et al., 2024b), molecular graph description (Fang et al., 2024), scene graph generation (Krishna et al., 2017), and event relation extraction (Wang et al., 2022; Glavas et al., 2014). Each instance is treated as a *graph state* represented by a set of relational tuples. For this analysis, we abstract away task semantics and focus purely on topology-level structure; when required by the metric, directed graphs are converted to an undirected view.

### A.2. Structural Statistics

We report four topology-level statistics that characterize coarse structural properties shared across domains: (1) **AvgDeg**: the mean node degree per graph (computed on the undirected view), capturing *relational density*; (2) **ASPL**: the average shortest path length per graph (undirected), capturing *global connectivity scale*; (3) **TwoHop**: the average number of length-2 paths $(A \rightarrow B \rightarrow C)$ per graph, capturing the prevalence of *two-step compositional dependencies*; (4) **Star**: the average number of hub nodes (degree $\geq 3$) per graph, capturing *hub-centric* relational organization. All statistics are computed by first aggregating counts within each graph and then averaging over the dataset.

| Domain | AvgDeg ↑ | ASPL | TwoHop ↑ | Star ↑ |
|---|---|---|---|---|
| Graph Algorithm | 5.8068 | 2.0787 | 632.7410 | 16.1844 |
| Molecular Graph Description | 2.1125 | 6.1799 | 52.5278 | 11.9700 |
| Scene Graph Generation | 1.5111 | 1.4485 | 2.4198 | 0.7078 |
| Event Relation Extraction | 1.4739 | 1.3731 | 14.9518 | 0.8642 |

*Table 5.* Topology-level structural statistics across four domains. AvgDeg and ASPL are per-graph averages computed on undirected views. TwoHop and Star quantify the prevalence of two-hop chains and hub nodes, respectively. These statistics are reported only to establish the recurrence of coarse structural patterns across domains, rather than to compare magnitudes or evaluate models.

**Cross-domain observations.** Table 5 reveals two complementary patterns. First, **local structural motifs recur broadly across domains**: all four settings exhibit non-trivial two-hop dependencies and hub nodes, indicating that compositional relational structure is not confined to any single modality or task formulation. Second, **global structural scale varies in an interpretable manner**. Molecular graphs exhibit substantially larger ASPL, consistent with their chain-like or near-tree chemical backbones. Algorithmic graphs are denser and contain many more two-hop dependencies, reflecting larger graph sizes and higher branching factors. In contrast, event and scene graphs are comparatively compact (ASPL $\approx 1.4$) with similar relational density (AvgDeg $\approx 1.5$), suggesting that compact relational organization can arise across both textual (event-centric) and visual (scene-centric) sources.

### A.3. Qualitative Evidence of Shared Structural Constraints Across Tasks

To further clarify how graph structure functions as a reusable substrate across heterogeneous tasks, we present qualitative evidence showing that *structurally identical motifs not only recur across domains, but also encode closely aligned constraint roles*. Specifically, we pair instances from **event graphs** (text-derived) and **scene graphs** (vision-derived) that instantiate the same coarse structural templates, and show that these templates impose similar relational constraints despite differences in semantics and relation inventories.

Throughout this section, motif identity is defined purely at the level of *topology* rather than label semantics. Our goal is not to establish semantic equivalence, but to demonstrate that shared structural forms correspond to shared constraint interpretations across tasks.

**Two-hop chains ($A \to B \to C$).** We first examine two-hop chains, which represent the minimal form of compositional relational structure. Across both domains, all examples instantiate the same role-aligned structural template:

$$\boxed{A} \;\to\; \boxed{B} \;\to\; \boxed{C}$$

*Constraint role:* an intermediate state $B$ composes two relations, imposing a mediated dependency between a source $A$ and an outcome $C$.

| Event graph (text) | Scene graph (vision) | Aligned constraint role |
|---|---|---|
| $E7 : took\_place \to E3 : ridden \to E2 : remounted$ | $board \to fence \to zebra$ | Mediated dependency via an intermediate anchor |
| $E13 : received \to E3 : believed \to E31 : appeared$ | $board \to counter \to bowl$ | Composition of two relations through $B$ |
| $E10 : gained \to E31 : published \to E18 : avalanches$ | $engine \to train \to track$ | Two-step mediated relational path |

*Table 6.* Cross-domain two-hop chain pairs. Although event and scene graphs differ in semantics and relation types, both instantiate the same compositional constraint: an intermediate node mediates dependencies between a source and an outcome.

Across all pairs, the identity of the intermediate node $B$ differs in meaning (e.g., temporal anchoring in event graphs versus spatial mediation in scene graphs), yet its *structural role* remains invariant: it serves as a compositional bottleneck that constrains how two relations interact. This consistency highlights that two-hop chains encode similar constraint semantics across tasks.

**Hub / star motifs (degree $\geq 3$).** We next examine hub motifs, which capture cases where a single node participates in many relations. Across both domains, these instances instantiate the same star-like template:

$$\boxed{H} \;\leftrightarrow\; \{n_i\}_{i=1}^k, \quad k = \deg(H)$$

*Constraint role:* a shared anchor $H$ simultaneously constrains multiple relations, enforcing global consistency across dependent nodes. In both event and scene graphs, hub nodes function as structural anchors rather than task-specific artifacts. They concentrate relational

| Event graph | | | Scene graph | | | Aligned constraint role |
|---|---|---|---|---|---|---|
| Hub | Deg | Rel. type | Hub | Deg | Rel. type | |
| $E6 : received$ | 4 | temporal | *counter* | 3 | spatial/support | Central anchor for multiple relations |
| $E3 : ridden$ | 6 | temporal | *guy* | 5 | part-of/attribute | Shared reference point across dependents |
| $E31 : published$ | 20 | temporal | *train* | 6 | part-of/spatial | Shared anchor coordinating multiple relations |

*Table 7.* Cross-domain hub motif pairs with aligned constraint interpretations. Despite different relation semantics, hubs in both domains act as shared anchors that impose multi-relation consistency.

constraints around a central node, allowing multiple relations to be coordinated through a shared reference point. This role remains consistent even though the surrounding relations encode different semantics (temporal, spatial, or part-of).

Together, these paired examples demonstrate that recurring structural motifs across tasks encode not only similar topological patterns, but also closely aligned *constraint roles*. This observation supports the representation-centric view adopted in Section 2.1: graph structure operates as a reusable intermediate substrate at the level of relational organization, abstracting away from task- or modality-specific semantics while preserving constraint-level meaning.

## B. Concrete Instantiation of the Unified Graph Representation

This section provides a concrete instantiation of the shared graph representation described in Section 2.1. The goal of this instantiation is not to prescribe a canonical graph format, but to illustrate one practical realization of the structural form used in **G-Substrate**.

In our experiments, each graph is represented as a collection of uniquely identified entities and typed, directed relations defined over ordered pairs of entities. Optional attributes may be attached to entities or relations when required by specific tasks, but are treated as auxiliary annotations and do not modify the underlying relational topology. Structural identity is therefore determined solely by relational connectivity, which is the property that must remain stable for graphs to be reusable across tasks and functional roles.

This structural form is used uniformly across tasks without encoding task-specific semantics, execution procedures, or optimization objectives. The same graph may be consumed as an intermediate representation in graph understanding tasks or produced as an output in graph generation tasks. Differences between tasks are expressed through prompts, supervision signals, and evaluation protocols, rather than through modifications to the graph structure itself.

We emphasize that this instantiation represents only one possible realization of the shared graph representation. **G-Substrate** does not depend on any particular schema choice, serialization format, or internal encoding, as long as graphs conform to a consistent entity–relation structural form that enables reuse across tasks.

| Component | Field | Description |
|---|---|---|
| Entity | id | Unique entity identifier (e.g., E1, E2) |
| | type | Optional entity category or label |
| Relation | subject | Source entity identifier |
| | predicate | Typed relation label |
| | object | Target entity identifier |
| Attribute | key | Optional attribute name |
| | value | Attribute value |

*Table 8.* Minimal structural primitives used in one realization of the shared graph representation.

**Graph representations across domains.** Although instantiated in different domains, all graphs used in our experiments conform to the same entity–relation abstraction: uniquely identified entities connected by typed relations over ordered pairs, with optional auxiliary attributes.

| Domain | Entity Example | Relation Example | Structural Role |
|---|---|---|---|
| Graph Algorithmic | node id (e.g., 5, 17) | CONNECTED | Pure topology |
| Molecular Graph | atom (C, O, C_aromatic) | SINGLE-BOND, AROMATIC-BOND | Chemical interaction structure |
| Event Graph | event mention (*destroyed*, *displaced*) | BEFORE, AFTER | Temporal / causal dependency |
| Scene Graph | object / part (cat, box, eye) | ON, HAS, OF | Spatial / semantic interaction |

*Table 9.* Examples of how different domains instantiate the same entity–relation structural abstraction. Variation lies in label vocabularies and supervision, while the relational form remains consistent.

Across domains, variation lies in label vocabularies and supervision protocols rather than in structural form. Structural identity is determined solely by relational connectivity over entity identifiers, enabling graphs to be reused across tasks with different functional roles without structural translation. While G-Substrate does not rely on any specific representation choice for its validity, different realizations may induce different inductive biases and therefore lead to quantitative differences in downstream performance. We empirically study this effect in Section 3.

## C. Task Coverage and Framework Instantiation

This section summarizes how the **G-Substrate** framework is instantiated across different task settings. Rather than enumerating dataset-specific configurations, we focus on how graph representations are generated, understood, and reused across tasks under the structural substrate defined in the main text.

| Task Setting | Input Modality | Graph Role | Reuse Pattern |
|---|---|---|---|
| Scene Graph Generation | Image | Generation | – |
| Event Relation Extraction | Text | Generation | – |
| Molecular Graph Description | Structured Molecule Input | Unstanding | – |
| Graph Algorithm Task | Graph | Unstanding | Reuses graphs produced in scene graph generation and event relation extraction |
| Subgraph Retrieve Task | Graph | Unstanding | Reuses graphs produced in scene graph generation and event relation extraction |
| Cross-Modal Consistency | Image + Graph | Unstanding | Bidirectional reuse between perception and structure |

Across the task instantiations considered in this work, graphs satisfy the same structural admissibility constraints defined by the unified schema, allowing Graph generated in one setting to remain structurally compatible with others. Differences between tasks arise from how graphs are generated and understood, rather than from changes to the underlying structural constraints. For cross-modal consistency tasks, we additionally introduce controlled structural perturbations to a subset of the reused graph to construct negative examples. These perturbations modify relational connectivity while preserving surface-level elements, enabling the model to distinguish structurally coherent graphs from inconsistent ones. This design ensures that consistent supervision provides meaningful structural learning signals rather than relying solely on positive reuse cases.

## D. Training Paradigm Definitions

This section provides operational definitions of the training paradigms summarized in Section 3.2. All paradigms use the same backbone model, optimizer, training schedule, data mixture, and total training budget. They differ only in (i) the representational constraints applied to graph states and (ii) how graph states are exposed to functional roles during training.

**Naive single-task.** Each task is trained independently using its original task-specific graph representation and supervision objective. Training batches contain examples from a single task only, and graph states are constructed, optimized, and consumed exclusively in that task. Graphs are optimized under a single functional role, and no cross-role exposure occurs.

**Unified single-task.** Each task remains trained in isolation, but graphs are represented using the unified structural schema described in Section 2.2. Although the same structural admissibility constraints apply across tasks, graphs are never exposed outside their originating task. Learning signals remain task-specific, and graphs still operate under a single functional role, so no reuse pressure is present.

**Naive multi-task.** All tasks are jointly trained by sampling batches from multiple tasks under their native graph formats. Model parameters are updated across tasks, but graphs are still constructed and can be understood only in their originating tasks. Graphs are therefore optimized under task-specific roles, without structural alignment or cross-role reuse.

**Unified multi-task (schema only).** Tasks are jointly trained while graphs are expressed using the unified structural schema. This imposes a common set of structural admissibility constraints across tasks, aligning graph representations at the structural level. However, graphs remain tied to their task of origin and are not exposed to different functional roles. Structural compatibility is established, but no cross-role reuse occurs.

**Naive multi-task + interleave.** Interleaved role-based training is introduced: the graph produced under one task-role instantiation may be reused as inputs under another task-role instantiation. This exposes the same graph state to multiple functional roles during training. However, graphs retain their task-specific formats, and no unified structural admissibility constraint is imposed. Cross-role reuse occurs, but under heterogeneous structural conventions.

**G-Substrate (Unified + interleave).** Our full framework combines the unified structural schema with interleaved role-based training. Graphs satisfy a common set of structural admissibility constraints and are explicitly reused under multiple functional roles during training. Learning, therefore, applies consistent pressure toward graph representations that remain structurally compatible and reusable across heterogeneous task contexts.

# E. Hyperparameter Configuration

Table 10 summarizes the shared hyperparameter configuration used across all experiments, including task-isolated training, naive multi-task learning, and the proposed **G-Substrate** framework. All compared methods use identical model backbones, optimization settings, training budgets, and decoding configurations. All experiments are performed on a server with four NVIDIA A100 GPUs (40GB each). Fine-tuning is implemented using the LLaMA-Factory framework.

| Hyperparameter | Value |
|---|---|
| Backbone models | Qwen3-VL-2B |
| Finetuning type | Full-parameter training |
| Vision tower frozen | Yes |
| MM projector frozen | No |
| Language model frozen | No |
| Optimizer | AdamW ($\beta_1$=0.9, $\beta_2$=0.98) |
| Learning rate | $8 \times 10^{-6}$ |
| Weight decay | 0.01 |
| Learning rate schedule | Cosine decay |
| Warmup ratio | 0.10 |
| Max input length | 2048 tokens |
| Mixed precision | bfloat16 |
| Per-device batch size | 1 |
| Gradient accumulation steps | 32 |
| Effective batch size | 64 sequences |
| Training epochs | 2 |
| Max gradient norm | 1.0 |
| Random seed | 42 |
| Decoding strategy | Greedy decoding (temperature = 0) |

*Table 10.* Shared training configuration used across experiments unless otherwise specified.

# F. Detailed Experimental Results

This appendix reports detailed results under different training paradigms of our framework, providing per-task and per-dataset breakdowns that complement the main experimental findings.

The details from table 11 to table 14 results across domains reveal a consistent interaction between representation format and training

| Method | Connectivity | Cycle | Shortest Path | Matching | Overall |
|---|---|---|---|---|---|
| Naive single-task | 99.44 | 92.18 | 38.27 | 92.05 | 92.89 |
| Unified single-task | 97.80 | 94.72 | 37.14 | 85.98 | 90.43 |
| Naive multi-task | 99.71 | 94.72 | 41.27 | 92.21 | 93.01 |
| Unified multi-task | 98.09 | 96.19 | 45.02 | 94.23 | 93.85 |
| Naive multi-task + interleave | 98.27 | 93.86 | 43.83 | 91.92 | 93.24 |
| G-Substrate | 98.41 | 96.97 | 48.59 | 94.54 | 94.47 |

*Table 11.* Detailed results on graph algorithmic tasks.

| Method | BLEU-4 | ROUGE-L |
|---|---|---|
| Naive single-task | 48.59 | 66.65 |
| Unified single-task | 47.35 | 65.64 |
| Naive multi-task | 48.11 | 66.11 |
| Unified multi-task | 49.99 | 67.36 |
| Naive multi-task + interleave | 48.63 | 64.98 |
| G-Substrate | 51.53 | 68.47 |

*Table 12.* Detailed results on molecular graph description.

| Method | PCIs R@50 | PCIs mR@50 | SGCLs R@50 | SGCLs mR@50 |
|---|---|---|---|---|
| Naive single-task | 23.74 | 7.78 | 11.95 | 4.10 |
| Unified single-task | 22.43 | 5.84 | 10.43 | 4.01 |
| Naive multi-task | 24.68 | 7.49 | 13.57 | 4.00 |
| Unified multi-task | 24.71 | 7.00 | 13.78 | 4.80 |
| Naive multi-task + interleave | 24.02 | 7.64 | 13.24 | 4.30 |
| G-Substrate | 25.38 | 8.67 | 14.07 | 5.30 |

*Table 13.* Detailed results on scene graph generation.

| Method | MAVEN-ERE | | | | | | | | | HiEve | | |
| | Subevent | | | Temporal | | | Causal | | | | | |
| | P | R | F1 | P | R | F1 | P | R | F1 | P | R | F1 |
|---|---|---|---|---|---|---|---|---|---|---|---|---|
| Naive single-task | 43.44 | 42.27 | 39.65 | 39.05 | 55.22 | 41.60 | 31.45 | 30.87 | 27.70 | 15.10 | 30.95 | 17.10 |
| Unified single-task | 52.80 | 44.63 | 45.45 | 37.84 | 42.79 | 33.29 | 38.07 | 29.45 | 30.22 | 13.14 | 21.83 | 14.28 |
| Naive multi-task | 40.44 | 39.27 | 36.87 | 37.05 | 53.22 | 39.14 | 37.45 | 37.87 | 37.02 | 15.29 | 30.41 | 18.78 |
| Unified multi-task | 56.36 | 53.52 | 51.89 | 44.74 | 48.84 | 40.05 | 45.41 | 43.53 | 40.75 | 18.34 | 25.38 | 19.37 |
| Naive multi-task + interleave | 50.43 | 47.32 | 45.74 | 39.41 | 42.89 | 38.86 | 41.44 | 43.24 | 37.99 | 20.43 | 24.74 | 21.36 |
| G-Substrate | 58.91 | 52.18 | 52.20 | 46.98 | 51.69 | 42.68 | 48.89 | 40.09 | 40.91 | 22.25 | 35.74 | 25.15 |

*Table 14.* Event relation extraction results on MAVEN-ERE and HiEve.

organization. In task-isolated settings, enforcing the unified schema alone does not yield gains and can even reduce performance, as structural constraints are not exercised beyond a single objective. Under multi-task learning, however, unified representations become beneficial, indicating that structural alignment matters once graph states are exposed to multiple learning contexts. The full G-Substrate framework further improves over both naive and unified multi-task baselines, with the largest gains observed in tasks that rely on multi-step relational composition, such as shortest-path reasoning, rare scene-graph relations, and event substructure modeling. By contrast, naive interleaving without schema-level alignment provides only limited and unstable improvements. These patterns suggest that performance gains arise not from task mixing alone, but from organizing learning so that structurally admissible graph states are reused across heterogeneous roles.

## G. Generality Across Model Backbones

To examine whether the observed performance gains are specific to a particular vision–language model backbone, we conduct a lightweight transfer study using an alternative vision–language model backbone from a different model family. This analysis is intended as a robustness check rather than an exhaustive model comparison.

We repeat a subset of the main experiments using InternVL3_5-2B-HF under the same training recipe, data composition, and evaluation protocol as in the main paper. Specifically, we compare task-isolated training, naive multi-task learning, and the full **G-Substrate** framework, along with key component ablations.

| Method | GAR | | | | MGD | | SGG | ERE | | | |
| --- | --- | --- | --- | --- | --- | --- | --- | --- | --- | --- | --- |
| | CT | CD | SP | BM | BLEU-4 | ROUGE-L | PCIs | MA-S | MA-T | MA-C | HiE |
| Naive single-task | 99.65 | 94.22 | 37.33 | 92.36 | 50.37 | 67.60 | 26.94 | 41.68 | 42.80 | 32.87 | 16.70 |
| Unified single-task | 97.86 | 94.29 | 36.96 | 92.36 | 49.89 | 67.44 | 28.15 | 52.65 | 37.82 | 34.80 | 10.74 |
| Naive multi-task | **99.84** | 95.42 | 45.81 | 92.67 | 51.45 | 68.28 | 26.59 | 43.40 | 43.84 | 32.47 | 18.22 |
| Unified multi-task | 98.26 | 95.98 | 47.46 | 94.39 | 51.77 | 68.77 | 28.29 | **57.04** | **45.62** | **41.57** | 18.78 |
| Naive multi-task + interleave | 98.04 | 94.86 | 46.89 | 93.48 | 51.63 | 67.98 | 28.02 | 53.74 | 38.94 | 38.74 | 21.36 |
| **G-Substrate** | 98.28 | **98.28** | **50.65** | **94.54** | **52.02** | **68.95** | **29.01** | 55.23 | 43.04 | 39.05 | **21.68** |

*Table 15.* Task-level performance using an alternative model backbone (InternVL). The same training recipe and evaluation metrics as in the main experiments are used. Best results are in **bold**; second-best are underlined.

Table 15 shows that the overall trends observed in the main experiments persist under a different model backbone. In task-isolated settings, the unified schema alone does not consistently outperform native representations, and in some cases slightly reduces performance, mirroring the behavior observed with the primary backbone. This again indicates that structural alignment by itself does not constitute an intrinsic performance advantage. Under multi-task training, however, unified representations become more effective. Unified multi-task learning improves over naive multi-task training across most domains, particularly in shortest-path reasoning (SP), scene graph metrics, and event relation extraction. The full G-Substrate framework further improves over both baselines, yielding the strongest or near-strongest results in most settings. Notably, gains are most visible in tasks that require multi-step relational composition, such as SP in GAR and subevent/causal relations in ERE, which are structurally similar to the patterns seen with the original backbone. Naive multi-task with interleaving provides partial benefits but remains less stable across domains, especially in ERE, where some metrics degrade relative to unified multi-task training. This again suggests that cross-task exposure alone is insufficient, and that consistent structural admissibility plays an important role in enabling reliable reuse.

Overall, the consistency of these patterns across two architecturally distinct vision–language backbones indicate that the improvements are not tied to a specific model family. Instead, they stem from how relational structure is represented and reused during training, supporting the generality of the framework.

# H. Extended Related Work

This appendix expands the discussion in Section 4 and situates our work within a broader landscape of research involving graph structure in learning systems. We organize prior work according to a common task formulations in which graphs arise, and analyze how graph representations are constructed, optimized, and used. Across these paradigms, a recurring pattern emerges: graph structure is typically introduced to satisfy the objective of an individual task, and is rarely maintained as a persistent intermediate representation that must remain compatible and reusable across tasks.

## H.1. Tasks over Structured Graphs with LLMs and VLMs

A substantial body of work has studied graph-structured data using graph neural networks and related graph representation learning methods. These approaches encode relational structure through message passing, neighborhood aggregation, and relation-aware propagation, and have been widely used to model typed entities, relations, and structured dependencies in graph data (Zhang et al., 2019a; Velickovic et al., 2018; Kipf & Welling, 2017). GNN-based methods have also been applied across diverse structured domains, including molecular modeling, knowledge-intensive reasoning, recommendation, and database systems (Li et al., 2025f; Schlichtkrull et al., 2018).

Recent LLM- and VLM-based methods broaden this line by studying tasks defined over structured graph inputs. These include graph-theoretic reasoning and algorithmic problems such as shortest path, connectivity, traversal, and combinatorial queries, often realized through graph serialization, specialized prompting, or graph-aware tokenization strategies (Li et al., 2025b; Wang et al., 2025b; Hu et al., 2025c; Chen et al., 2024b; Wang et al., 2024b; 2023; Yuan et al., 2025b; Zhang et al., 2024b; Tang et al., 2024b; Wang et al., 2025a). More recent work extends such settings to multimodal regimes, where graphs are derived from images or other perceptual signals and processed by VLMs (Wei et al., 2024; Sartori et al., 2025; Li et al., 2024b; Zhu et al., 2025; Zhao et al., 2025b;d). Related efforts address tasks such as molecular graph description and reasoning, where structured graphs are mapped to semantic outputs (Kim et al., 2025; Liu et al., 2024c; Park et al., 2024; Fang et al., 2024; Yin et al., 2025; Luo et al., 2024b; Wu et al., 2025b; Jin et al., 2025).

Across these works, the graph is typically treated as a task-bounded input object. Its representation is optimized only insofar as it supports the current objective (e.g., algorithmic prediction or description generation). There is no requirement that graph representations remain structurally compatible with other tasks, nor that they serve as intermediate artifacts reused under different learning roles.

## H.2. Graph Generation in Vision and Language

Another major line of work focuses on generating graphs from perceptual or linguistic inputs, such as scene graph generation from images (Chen et al., 2024d; Li et al., 2024a; Liu et al., 2025; Xu et al., 2025c; Hu et al., 2025b; Min et al., 2025; Hu et al., 2025a; Wang et al., 2025c; Elskhawy et al., 2025; Chen et al., 2025b; Dutta et al., 2025; Hu et al., 2025d; Kong & Zhang, 2025; Li et al., 2025e) and

event–event relation extraction from text (Hu et al., 2025e; Xu et al., 2025b; Chen et al., 2024b; Ding et al., 2025; Zhao et al., 2025c; Tanev et al., 2025; Li et al., 2025a; Wang et al., 2024d; Liu & Wang, 2025; Chen et al., 2024a). In these formulations, graphs serve as final prediction targets. Training objectives optimize graph quality with respect to task-specific metrics, and the Generated graphs are evaluated independently within each task context.

As a result, graphs are not required to persist beyond generation or to function as reusable intermediate representations for other tasks. Structural regularities learned during generation are not explicitly constrained to remain compatible with graph-conditioned reasoning tasks.

### H.3. Multi-Task and Multi-Modal Learning

Multi-task and multi-modal learning have been extensively studied as mechanisms for coordinating learning across tasks and modalities (Ruder, 2017; Akhtar et al., 2020; Yuan et al., 2025a; Zhang & Yang, 2022). Typical approaches emphasize parameter sharing (Pan et al., 2025; Liu et al., 2024a; Leng & Xiong, 2025), task balancing (Xia et al., 2024; Zhao et al., 2025a; Gong et al., 2024; Ju et al., 2023), curriculum design (Chen et al., 2025a; Zhao et al., 2025a; Wang et al., 2024c), and optimization heuristics (Xia et al., 2024; Zhang et al., 2025; Leng & Xiong, 2025). Recent work has also explored broader forms of foundation-model specialization, including domain-specific VLM (Ma et al., 2025a) and skill-graph-based data selection for mathematical pretraining (Li et al., 2025d).

These methods coordinate learning primarily at the level of parameters, losses, data scheduling, or task curricula. When graph structure appears, it usually serves as a task-specific input, output, or data-organization signal, and reuse occurs implicitly through shared parameters rather than through explicit reuse of intermediate graph states. In contrast, our framework treats graph states as persistent intermediate representations that must remain structurally valid across heterogeneous generation and understanding roles.

### H.4. Unified and Foundation Graph Models

Graph foundation models aim to build general-purpose systems that transfer across graph tasks and domains through large-scale pretraining, architectural unification, and broad task coverage. Existing approaches can be broadly categorized into three directions: *graph-model–centric* methods that extend graph neural architectures toward broader generality (Liu et al., 2024b; Wang et al., 2024f;e; Yu et al., 2024; Jiang et al., 2024); *language-model–centric* methods that adapt LLMs to operate on graph-structured inputs or tasks (Li et al., 2025c; Lin et al., 2024; Kong et al., 2025; Wang et al., 2024b; 2025f); and *joint graph–language pretraining* approaches that co-train graph and language representations within a unified frameworks (Tang et al., 2024a; Luo et al., 2024a; Chen et al., 2024c; Zhang et al., 2024a; Liu et al., 2024d; Wang et al., 2024a; Hu et al., 2024; Wang et al., 2025e; Liu et al., 2023; Wang et al., 2025d; Thapaliya et al., 2025; Ma et al., 2025b). These models emphasize scale, pretraining diversity, and architectural unification, aiming to improve transfer across graph tasks through shared parameters and large training corpora. However, the graph structure in these systems remains conditioned on task formulations: graph representations are constructed and optimized with respect to individual task objectives, and are not required to persist as intermediate artifacts beyond the originating task.

Our work explores a complementary axis of generalization. Rather than focusing On how parameters or model architectures generalize across tasks, we study how *intermediate graph states themselves* can be organized to remain structurally admissible and reusable under heterogeneous task roles. We explicitly enforce structural compatibility and cross-task reuse of the graph states, treating graphs as a reusable substrate in the learning process rather than as task-bound artifacts. This perspective is orthogonal to scaling and architectural unification, and addresses how structured representations persist and function across learning contexts.

## I. Comparison with Gradient-Balancing Multi-Task Baselines

To further situate G-Substrate relative to standard multi-task learning algorithms that address task interference at the optimization level, we compare against **GradNorm** (Chen et al., 2018), a widely used gradient-balancing method. GradNorm dynamically reweights task losses to equalize gradient magnitudes across tasks, and we apply it on top of the naive multi-task baseline (NMT). For fair comparison, we also evaluate G-Substrate combined with GradNorm. We use $\alpha = 1.5$ and a weight learning rate of $0.025$, following the original recipe.

*Table 16.* **Comparison with gradient-balancing multi-task learning.** We report averaged accuracy for GAR, BLEU-4 for MGD, PCIs R@50 for SGG, and macro-averaged F1 for ERE.

| Method | GAR | MGD | SGG | ERE |
|---|---|---|---|---|
| NMT | 93.01 | 48.11 | 25.36 | 38.02 |
| NMT + GradNorm | 93.43 | 49.64 | 25.35 | 35.09 |
| G-Substrate | 94.47 | 51.53 | 25.38 | 42.24 |
| G-Substrate + GradNorm | 94.39 | 52.49 | 26.41 | 40.24 |

Two observations follow. First, GradNorm improves NMT on GAR and MGD but *hurts* ERE by $-2.93$, because gradient-magnitude equalization assigns near-zero weight to event-relation extraction once that loss converges faster. This illustrates a known limitation of convergence-based reweighting under heterogeneous task difficulty. Second, G-Substrate outperforms NMT+GradNorm on *all* four domains *without any gradient balancing*, indicating that the dominant bottleneck in our setting is *representational*—how relational structure is shared and reused—rather than optimization-level loss balancing. Combining G-Substrate with GradNorm produces mixed effects (MGD $+0.96$, SGG $+1.03$, ERE $-1.98$), suggesting that convergence-based reweighting can interfere with the balanced cross-role

exposure that G-Substrate relies on. The two approaches address complementary but distinct bottlenecks, and gradient balancing is not a substitute for explicit representation reuse.

## J. Robustness to Noisy Graph Extraction

In practical pipelines, graph extraction is rarely perfect: scene graph generators, event extractors, and parsers all produce structurally imperfect graphs. To evaluate whether G-Substrate's gains depend on access to clean graphs, we simulate imperfect extraction by injecting controlled structural noise into the graphs reused during interleaved training. At each noise level, a fixed proportion of edges is randomly perturbed through a mixture of operations: relation-label replacement (40%), entity substitution (30%), subject–object swapping (15%), and edge deletion (15%). Noise is applied only to graphs used as interleaved cross-role training data; the primary task data remains unperturbed.

*Table 17.* **Robustness of G-Substrate under noisy graph extraction.** Performance on the four domains as the proportion of perturbed edges increases from 0% (clean G-Substrate) to 30%. NMT (clean) is provided as a reference. We report averaged accuracy for GAR, BLEU-4 for MGD, PCIs R@50 for SGG, and macro-averaged F1 for ERE.

| Noise Level | GAR | MGD | SGG | ERE |
|---|---|---|---|---|
| 0% (G-Substrate) | 94.47 | 51.53 | 25.38 | 42.24 |
| 10% | 94.73 | 49.09 | 24.59 | 41.33 |
| 20% | 92.10 | 50.74 | 23.29 | 39.74 |
| 30% | 92.24 | 47.93 | 23.81 | 37.60 |
| NMT (clean) | 93.01 | 48.11 | 25.36 | 38.02 |

Performance degrades gradually with *no catastrophic failure*. G-Substrate under 20% noise still outperforms clean NMT on MGD (50.74 vs. 48.11) and ERE (39.74 vs. 38.02), and remains competitive on GAR (92.10 vs. 93.01). SGG is more sensitive to noise, dropping below clean NMT at all noise levels, likely because scene graphs are structurally compact (average 1.5 edges per relation, Table 1) and thus more affected by per-edge perturbation. Even at 30% noise, three of four domains remain close to or above clean NMT levels.

This result is consistent with Figure 5: complete structural corruption reverses gains, but partial noise leads to graceful degradation, indicating that G-Substrate does not require perfect graph extraction to remain effective. This robustness likely arises because cross-role reuse acts as a structural regularizer: only structurally consistent patterns shared across tasks are reinforced, so noise in any single context does not dominate the learned representation.

