# OpenReview forum: "Graph is a Substrate Across Data Modalities"
_ICML.cc/2026/Conference — ICML 2026 regular_

### Official Review · Reviewer_rxB1 · 2026-03-10

**Soundness:** 3
**Presentation:** 3
**Significance:** 3
**Originality:** 3
**Overall Recommendation:** 5
**Confidence:** 4

**Summary:**

The authors develop a new model that uses graph structure as a reusable intermediate substrate across modalities and tasks. The proposed method promotes structural compatibility and cross-task reuse for different domains. Experiments on four domains demonstrate consistent gains over task-isolated and standard multi-task baseline methods. Overall, the idea is conceptually new and interesting, and the empirical evidence is generally consistent with the proposed method.

**Compliance With Llm Reviewing Policy:**

Affirmed.

**Final Justification:**

All my concerns have been addressed. I will keep my positive score.

**Key Questions For Authors:**

As mentioned in the weaknesses, please show the performance in the case of noisy graph extraction and demonstrate the complexity-effectiveness trade-off of the proposed method.

**Limitations:**

How modality, domain, or role composition shapes representations, and the optimal balance among generation, understanding, and algorithmic roles, remains unclear.

**Strengths And Weaknesses:**

Strengths:
1. The idea of treating graph structure as a reusable intermediate substrate across domains distinguishes itself from prior multi-task and graph foundation model work. It generalizes to multiple domains and data modalities. It is a well-motivated and new idea.
2. The experimental design is carefully controlled. By constructing multiple training paradigms that isolate schema alignment and role interleaving effects, the authors avoid conflating representation design with multitask scaling.
3. The evaluation spans graph algorithms, molecular graph description, scene graph generation, and event relation extraction, which is solid.

Weaknesses:
1. The approach presumes accurate graph construction. In practical pipelines, graph extraction can be noisy. The robustness of persistent graph reuse under imperfect generation is not fully explored.
2. Although improvements are consistent, the added training complexity may not always justify the performance gains, depending on the application context.

---

> ### Author Rebuttal · Authors · 2026-03-30
>
> We thank the reviewer for the thoughtful questions and suggestions.
> ___
> For **W1 and Q1**, to simulate imperfect graph extraction in practical pipelines, we inject controlled structural noise into the graphs reused during interleaved training. At each noise level (10%, 20%, 30%), a corresponding proportion of edges are randomly perturbed through a mixture of operations: replacing relation labels (40%), substituting entities (30%), swapping subject and object (15%), and deleting edges (15%). Noise is applied only to the interleaved cross-role training data; the primary task data remains unperturbed.
>
> | Noise Level | GAR | MGD | SGG | ERE |
> |---|---|---|---|---|
> | 0% (G-Substrate) | 94.47 | 51.53 | 25.38 | 42.24 |
> | 10% | 94.73 | 49.09 | 24.59 | 41.33 |
> | 20% | 92.10 | 50.74 | 23.29 | 39.74 |
> | 30% | 92.24 | 47.93 | 23.81 | 37.60 |
> | NMT (clean) | 93.01 | 48.11 | 25.36 | 38.02 |
>
> Performance degrades gradually with **no catastrophic failure**. G-Substrate under 20% noise still outperforms clean NMT on MGD (50.74 vs. 48.11) and ERE (39.74 vs. 38.02), and remains competitive on GAR (92.10 vs. 93.01). SGG is more sensitive to noise, dropping below clean NMT at all noise levels, likely because scene graphs are structurally compact (average 1.5 edges per relation, Table 1) and thus more affected by per-edge perturbation. Even at 30% noise, three of four domains remain close to or above clean NMT levels. This result is consistent with Figure 5: complete structural corruption reverses gains, but **partial noise leads to graceful degradation**, indicating that G-Substrate does not require perfect graph extraction to remain effective. This robustness likely arises because **cross-role reuse acts as a structural regularizer**: only structurally consistent patterns shared across tasks are reinforced, so noise in any single context does not dominate the learned representation.
> ___
> For **W2**, G-Substrate **reduces overall complexity** compared to task-isolated pipelines: **one unified model replaces four separate models**, lowering total training and maintenance cost. G-Substrate introduces **no additional parameters or architectural components** compared to multi-task baselines; the only difference is how training data is organized (unified schema + interleaved roles). The improvements therefore come at **no additional computational cost** relative to standard multi-task training.
> ___
> For **L1**, we agree this is an important direction. Our current work provides partial evidence on how composition shapes representations:
>
> **Role composition (Figure 4):** Different cross-role supervision types contribute differently depending on domain structural characteristics. Highly constrained graph domains (e.g., GAR: 94.2 → 94.5) show smaller marginal gains, whereas more weakly constrained domains (e.g., ERE: 38.0 → 40.2) benefit more.
>
> **Domain composition (Table 4):** Adding event-centric graphs to training improves scene graph generation by **+2.37 PCIs R@50** (19.10 → 21.47), demonstrating that cross-domain composition directly shapes the learned representations even without target-domain data.
>
> **Interleaving proportion (Figure 6):** Moderate levels (50%) yield the best results across all domains, while excessive role-based exposure (100%) degrades performance.
>
> These results provide initial evidence that **composition effects are systematic and domain-dependent**.

---

> > ### Author Rebuttal · Reviewer_rxB1 · 2026-04-02
> >
> > Thanks for the author's response. All my concerns have been addressed. I will keep my positive score.

---

> > > ### Author Response · Authors · 2026-04-04
> > >
> > > We are grateful for the reviewer's continued engagement and valuable suggestions, which have helped strengthen the paper. We are glad to see that most concerns have been resolved, and we will include all supplementary experiments in the revised manuscript.

---

### Official Review · Reviewer_jNPX · 2026-03-12

**Soundness:** 4
**Presentation:** 3
**Significance:** 3
**Originality:** 3
**Overall Recommendation:** 5
**Confidence:** 4

**Summary:**

In this paper, the authors study how to organize graph structure across heterogeneous modalities and tasks and propose G-Substrate, a novel framework that treats graph structure as a persistent intermediate substrate across contexts. Specifically, it combines a unified structural schema with interleaved role-based training, enabling structural reuse across generation and understanding tasks. Extensive experiments across graph algorithmic reasoning, molecular description, scene graph generation, and event relation extraction demonstrate consistent improvements over baseline methods.

**Compliance With Llm Reviewing Policy:**

Affirmed.

**Final Justification:**

The authors‘ responses have addressed my comments. I maintain my positive score. I believe this paper makes a solid contribution towards multimodal graph foundation model.

**Key Questions For Authors:**

Besides the weaknesses mentioned above, I also want to ask the authors whether the proposed framework can be extended beyond entity–relation graphs to hypergraphs or higher-order relational structures without modifying the training framework?

**Limitations:**

Yes

**Strengths And Weaknesses:**

Strengths:
S1. The proposed method forms a novel perspective, i.e., G-Substrate takes graphs as persistent structural substrates, shifting focus from parameter sharing to intermediate representation reuse across modalities and functional roles.
S2. The proposed method provides a valid framework that unifies graph structure across multiple domains and data (e.g., graph, text, and image). It might be useful for tasks in different applications and inspire future work in related domains.
S3. Experiments are comprehensive. The model demonstrates strong performance across four heterogeneous domains (graph reasoning, molecular description, scene graph generation, event extraction), with consistent improvements over many baseline methods.
S4. The paper is easy to follow, with clear organization and good presentation.

Weaknesses:
W1. The framework assumes access to explicitly constructed graph representations, while its applicability to tasks where structure is not explicit or must be implicitly inferred remains unclear.
W2. It is unclear about the sensitivity to graph size, and it would be better to investigate the scalability to larger graph representations.

---

> ### Author Rebuttal · Authors · 2026-03-30
>
> We thank the reviewer for the constructive feedback.
> ___
> For **W1**, G-Substrate **deliberately focuses on tasks with explicit or extractable relational structure**. This is a principled design choice that ensures the structural patterns shared across tasks carry genuine relational semantics. Introducing arbitrary graph constructions (e.g., fully-connected graphs over token sequences) would dilute the signal that enables cross-task structural reuse.
> Tasks with explicit graph structure are **pervasive across ML**: knowledge graphs, molecular graphs, scene graphs, event relation graphs, program dependency graphs, circuit netlists, social networks, among others. Our four evaluation domains, spanning vision, language, scientific and algorithmic reasoning, are representative of this broad class, not exhaustive.
>
> For tasks with implicit structure, existing tools such as dependency parsing, AMR parsing, or information extraction pipelines can produce explicit graph representations that are directly compatible with G-Substrate.
> ___
> For **W2**, we note that G-Substrate **does not introduce additional scalability overhead**: graphs are represented as sequences of triplets, so complexity scales linearly with the number of edges, identical to the NMT baseline. We provide a graph-size sensitivity analysis, grouping test instances by graph size (number of edges) within each domain and reporting G-Substrate and NMT performance within each bucket.
>
> | Domain | Size Bucket | G-Substrate | NMT | Δ |
> |---|---|---|---|---|
> | GAR | Small (≤32) | 90.00 | 75.47 | +14.53 |
> | GAR | Medium (33–78) | 95.72 | 93.26 | +2.46 |
> | GAR | Large (>78) | 97.18 | 97.42 | -0.24 |
> | MGD | Small (≤27) | 48.08 | 43.52 | +4.56 |
> | MGD | Medium (28–39) | 52.40 | 53.79 | -1.39 |
> | MGD | Large (>39) | 58.41 | 47.02 | +11.39 |
> | SSG | Small (≤3) | 24.58 | 24.71 | -0.13 |
> | SSG | Medium (4–5) | 25.69 | 24.44 | +1.25 |
> | SSG | Large (>5) | 24.28 | 24.32 | -0.04 |
> | ERE | Small (≤4) | 47.26 | 40.74 | +6.52 |
> | ERE | Medium (5–7) | 49.60 | 39.59 | +10.01 |
> | ERE | Large (>7) | 45.05 | 36.55 | +8.50 |
>
> G-Substrate maintains consistent improvements across most size buckets and domains, with **no systematic degradation as graph size increases**. Two patterns emerge. First, G-Substrate shows the **largest gains on structurally complex graphs** (e.g., Molecule large: +11.39, Event Graph large: +8.50), suggesting that cross-task structural reuse becomes more valuable as relational complexity increases. Second, in Graph Reasoning, the largest gain appears on **small graphs (+14.53)**, where structural information is sparse and task-isolated training struggles (NMT: 75.47%). In both cases, the underlying mechanism is consistent: **G-Substrate provides the greatest benefit where structural information from a single task is insufficient**, whether due to graph complexity or graph sparsity. In a few individual buckets, G-Substrate shows marginal drops (e.g., Graph Reasoning large: -0.24, Scene Graph small: -0.13), but these are small in absolute magnitude and do not indicate systematic degradation. The only non-trivial drop (Molecule medium: -1.39) is offset by the strongest gain in the same domain (Molecule large: +11.39), confirming that the overall benefit within each domain remains positive.
> ___
> For **Q1**, **yes**. The structural triplet representation (u, r, v) can accommodate higher-order relations through standard reification techniques:
> - **Hyperedge:** An n-ary relation R(v1, v2, ..., vn) is represented by introducing an auxiliary node h_R and binary edges: (v1, role_1, h_R), (v2, role_2, h_R), ..., (vn, role_n, h_R).
> - **Higher-order relations:** Nested or hierarchical relations can be similarly decomposed into collections of binary triplets with auxiliary nodes.
>
> Since the reified result remains a set of (u, r, v) triplets, it naturally resides in $G_s$. **The training framework (interleaved role-based training) requires no modification.**

---

> > ### Author Rebuttal · Reviewer_jNPX · 2026-04-02
> >
> > The authors‘responses’ have addressed my comments. I maintain my positive score.

---

> > > ### Author Response · Authors · 2026-04-04
> > >
> > > We sincerely appreciate the reviewer's thoughtful and constructive engagement throughout the review process. We are encouraged that our responses have addressed the majority of the concerns raised, and we will integrate the supplementary experiments into the revised manuscript.

---

### Official Review · Reviewer_chNm · 2026-03-13

**Soundness:** 2
**Presentation:** 2
**Significance:** 3
**Originality:** 3
**Overall Recommendation:** 4
**Confidence:** 4

**Summary:**

The authors argue that graphs should be regarded as a structural substrate connecting various modalities and learning tasks. To this end, they propose G-Substrate based on two mechanisms: a unified structural schema and interleaved role-based training.

**Compliance With Llm Reviewing Policy:**

Affirmed.

**Final Justification:**

I found this paper interesting and thought-provoking, and I believe it puts forward a genuinely original perspective that could be valuable to the community. The idea of viewing graphs as a shared structural substrate across tasks and modalities is a compelling one, even though I still have some concerns about the strength of the empirical support and the clarity of a few broader claims. In particular, I can see this work opening up useful follow-up directions, both conceptually and methodologically. The rebuttal helped clarify the authors’ intent and reinforced my view that the contribution is broader than a narrow empirical improvement. Overall, while the paper still has some weaknesses, I came away feeling positive about its originality and its potential to spark further work, which is why I settled on a weak accept recommendation.

**Key Questions For Authors:**

- On page 3, the authors claim that a graph is defined by structural triplets that exclude task-specific semantics. However, if we consider examples such as (A, married, B) versus (A, drives, C), doesn’t the relation itself carry semantic meaning?
- In Section 2.1, G is used to denote a generic graph. Later, in Section 2.2, Gₛ is introduced to denote a unified graph state space that includes all such generic graphs G. It is unclear what this construction achieves if it contains every possible graph. This seems analogous to referring to the space of all probability distributions, which only becomes useful once we restrict it to a structured family (e.g., the exponential family).
- How is convergence ensured when using the proposed interleaving strategy?
- The paper states that graphs that are not compatible with all tasks are disfavored. Is it possible that no graph satisfies this requirement? Alternatively, could the final solution collapse to an overly simple graph that trivially satisfies all tasks?

**Limitations:**

Overall, I think the manuscript provides an interesting conceptual framework; however, it fails to establish its usefulness in practice.

**Strengths And Weaknesses:**

Strengths
- The core idea that graphs provide a powerful abstraction of relationships and that they may be used to connect modalities at an abstract level is reasonable and interesting.
- I found the paper conceptually interesting.

Weaknesses
- I am not sure what the authors mean by “a graph is built to serve a single objective and discarded after training.” While graphs may be defined with respect to a specific task, they are typically not discarded after training; rather, they are used during inference to examine relationships and support future predictions.
- The motivating example in Section 2 is very vague. In Table 1, the authors present summary statistics of topological properties across domains and claim that these indicate cross-domain invariance at the level of graph structure. It is unclear how this conclusion follows from Table 1. Moreover, it is not clear how such a conclusion could be generalized to support their broader claim about graphs in general.
- I did not find the results convincing. Related methods are discussed only after the results are presented. As a result, the authors avoid benchmarking their approach against existing multi-task learning methods. Even if those methods do not explicitly focus on identifying recurring graph structures, they should still be included in the experimental comparison.

---

> ### Author Rebuttal · Authors · 2026-03-30
>
> We thank the reviewer for the detailed and constructive feedback.
> ___
> For **W1**, we acknowledge that our phrasing could be clearer. By "discarded," we mean that in task-isolated pipelines, the graph representation constructed during one task's training is **not reused by other tasks**. Each task reconstructs graph structure from scratch, so structural regularities are **repeatedly rediscovered rather than accumulated**. We will revise the wording to "not reused across tasks" to avoid this ambiguity.
> ___
> For **W2**, Table 1 and Figure 2 serve complementary roles: Table 1 shows local motifs (e.g., hubs) recur across domains; Figure 2 shows these motifs play **analogous functional roles** in both event graphs (temporal) and scene graphs (spatial), despite different semantics. We will revise the presentation to avoid overstating Table 1. **The conclusive evidence is empirical**: Table 4 shows training on event graphs alone improves scene graph generation by **+2.37 PCIs R@50** (19.10 → 21.47) without target-domain supervision. Figure 5 confirms transfer depends on structural coherence. **We do not claim generality over all graphs**, but over tasks with explicit relational structure.
> ___
> For **W3**, Table 2 already includes standard multi-task learning baselines (NMT, UMT, NMT-I). To further address this concern, we compare with GradNorm(α=1.5, weight learning rate=0.025).
>
> | Method | GAR | MGD | SGG | ERE |
> |---|---|---|---|---|
> | NMT | 93.01 | 48.11 | 25.36 | 38.02 |
> | NMT + GradNorm | 93.43 | 49.64 | 25.35 | 35.09 |
> | G-Substrate | 94.47 | 51.53 | 25.38 | 42.24 |
> | G-Substrate + GradNorm | 94.39 | 52.49 | 26.41 | 40.24 |
>
> NMT + GradNorm improves GAR and MGD but **hurts ERE (-2.93)**, because GradNorm assigns near-zero weight to event graphs which converge faster in loss. **G-Substrate outperforms NMT + GradNorm on all four domains without any gradient balancing**, confirming **the bottleneck is representational, not optimization-level**. G-Substrate + GradNorm improves MGD (+0.96) and SGG (+1.03) but hurts ERE (-1.98), suggesting convergence-based reweighting can conflict with balanced cross-role exposure. This confirms the two approaches address different bottlenecks. Regarding the placement of Related Work, this follows a common convention at ICML.
> ___
> For **Q1**, relation labels like "married" and "drives" do carry semantic meaning, and **we do not strip them away**. They are fully preserved as typed edges in $G_s$. What we exclude is **task-specific semantics**: the same graph can serve as the output of one task (e.g., scene graph generation) and the input of another (e.g., graph reasoning) without any structural modification. Table 4 confirms this: training on event-centric graphs (relations like "before/overlap/cause") improves scene graph generation (relations like "on/near/has") by **+2.37 PCIs R@50, despite completely different relation semantics.**
> ___
> For **Q2**, the reviewer's analogy is apt. **$G_s$ does not contain every possible graph.** It is precisely the "structured family" the reviewer describes, constrained by the unified schema: consistent node identifiers across tasks, typed edges following fixed conventions, and a structural triplet format (u, r, v). Graphs that do not conform to these conventions are excluded from $G_s$. Table 3 confirms this empirically: under identical training, alternative schema realizations yield consistently lower performance (e.g., SP: Natural Language 44.80, XML-style 40.10, Ours **48.59**; HiE: 23.90, 20.80, **25.15**). **If $G_s$ were a trivial universal set, the choice of realization would not matter.**
> ___
> For **Q3**, interleaved role-based training is a **training organization strategy, not an iterative fixed-point procedure**, requiring no separate convergence guarantee beyond standard SGD convergence.
> Regarding potential interference: generation and understanding impose **complementary rather than competing pressures**. Figure 6 confirms stability: performance varies smoothly with interleaving proportion (0%, 50%, 100%) with **no oscillation** (e.g., GAR: 93.8→94.5→93.4, ERE: 38.0→40.2→38.4). If the two roles conflicted, we would expect unstable behavior rather than this consistent pattern.
> ___
> For **Q4**, this concern would arise if compatibility required graphs to be identical across tasks. **It does not.** $G_s$ imposes compatibility at the **structural format level, not the content level**. Each task produces its own graph with entirely different entities and relations. What they share is the same (u, r, v) format and structural patterns. **There is no requirement for a single graph to satisfy all tasks simultaneously.**
> Regarding collapse: a trivially simple graph would incur high generation loss and high understanding loss. Figure 5 confirms this: disrupting relational connectivity causes **sharp performance drops** (ERE: +2.23 vs. −2.03), confirming that training **actively maintains structural complexity**.

---

> > ### Author Rebuttal · Reviewer_chNm · 2026-04-02
> >
> > Thanks for the author's response. My concerns have been addressed. I adjusted my score accordingly

---

> > > ### Author Response · Authors · 2026-04-04
> > >
> > > We sincerely thank you for your positive feedback and helpful suggestions. We are glad to see that most of your concerns have been addressed. We will incorporate the supplementary experiments presented here into the revised manuscript.

---

### Official Review · Reviewer_i4vm · 2026-03-13

**Soundness:** 2
**Presentation:** 2
**Significance:** 2
**Originality:** 3
**Overall Recommendation:** 2
**Confidence:** 3

**Summary:**

This manuscript proposes G-substrate, a framework designed to handle a variety of graph-based tasks in a unified manner. G-substrate unifies graph-based tasks that were trained separately, such as scene graph generation and event graph generation, by introducing a single graph schema that maps different graphs into a unified graph space. Within this space, an internal graph is iteratively augmented and reused across multiple tasks. The proposed framework demonstrates better performance than task-specific models on several benchmarks, including graph algorithm reasoning, molecular graph description, scene graph generation, and event relation extraction.

**Compliance With Llm Reviewing Policy:**

Affirmed.

**Final Justification:**

After reading the rebuttal and reply comment, key concerns remain unresolved, including page limit violation, overstated claims about "unified graph state space", overclaimed contribution regarding graph usage, and insufficient justification of the design. Thus, I maintain my negative score.

**Key Questions For Authors:**

1. Can the framework be generally applied to tasks that do not explicitly utilize graph structures?
2. Why is cross-role reuse considered a necessary design requirement?
3. What does the unified schema look like in practice? Could the authors provide a concrete example illustrating $G^{(0)}$~$G^{(N)}$ ?

**Limitations:**

Yes. However, the limitations are described outside the page limit.

**Strengths And Weaknesses:**

## Strengths
1. The framework is applicable to a wide range of tasks that utilize graph structures.
2. A thorough component-wise analyses of the proposed model is provided.
3. The rationale for adopting a unified graph space is clearly explained.

## Weaknesses
1. The Limitations and Future Work paragraph is described on page 9. However, the ICML guidelines do not state that limitations or future work are exempt from the page limit. Therefore, this manuscript appears to violate the page limit and should be rejected on this basis.
2. The framework appears to be applicable only to tasks that operate on graph structures. For example, it does not seem applicable to tasks such as simple text generation that do not explicitly use graphs.
3. Overall, the explanation lacks sufficient detail. More concrete examples of the unified schema are needed, as well as a clearer description of how graphs are constructed and modified within the framework. Based on the current description in the main paper, it is difficult to understand how graphs are actually formed within the unified graph space.
4. Some design choices are insufficiently justified. While treating the graph structure as a persistent substrate is an interesting perspective, the paper does not explain why this perspective necessarily leads to the design requirements of structural compatibility and cross-role reuse. In particular, stronger justification is needed for the requirement of cross-role reuse, as well as discussion on whether additional design requirements might exist.

---

> ### Author Rebuttal · Authors · 2026-03-30
>
> We thank the reviewer for the detailed feedback. We address each point below.
> ___
>
> For **W1**, we clarify that this is not a page limit violation. The "Limitations and Future Work" paragraph is part of the Impact Statement in our paper, which, according to the ICML 2026 Call for Papers, **does not count toward the main 8-page limit**. Importantly, this paragraph contains no technical contributions, methods, or experimental details, and is included solely to discuss potential risks (e.g., domain-specific biases and cross-domain transfer issues) as required by the Impact Statement guidelines. **All technical content is strictly confined to the 8-page main body.** To remove any possible ambiguity, we will eliminate the separate heading and integrate this discussion directly into the Impact Statement section in the revision.
> ___
>
> For **W2 and Q1**, **our goal is not to generalize to all tasks, but to study whether relational structure itself can serve as a reusable intermediate state across tasks.** G-Substrate therefore focuses on settings where such structure is explicit or can be reliably extracted. Within this scope, graph-structured tasks are widespread across ML (e.g., knowledge graphs, molecules, scene graphs, event graphs, program graphs), and **our experiments span four modalities with distinct relation vocabularies, demonstrating generality at the level of structure rather than task or modality.** For tasks without explicit graphs, existing pipelines (e.g., dependency or AMR parsing) can make relational structure explicit. For instance, "The cat sat on the mat" can be converted via dependency parsing into a relational graph where "sat" connects to "cat" (nsubj) and "mat" (obl). **However, our contribution is not in constructing such graphs, but in organizing and reusing them as a persistent structural substrate.**
> ___
>
> For **W3 and Q3**, we illustrate the trajectory $G^{(0)}$ → $G^{(N)}$ concretely:
>
> **Step 0:** $G^{(0)}$ is an empty or minimal graph in the unified graph state space $G_s$.
>
> **Step 1 (GENERATE role):** Given an image of a street scene, the model produces $G^{(1)}$ = { (rider, on, horse), (horse, on, grass), (rider, wearing, helmet) }. For a different modality, an event extraction task produces $G^{(1)}$ = { (announced, before, visited), (visited, overlap, stated) }. **Both use the same structural format (u, r, v) with consistent node identifiers and typed edges.**
>
> **Step 2 (UNDERSTAND role):** A graph in the same unified format serves as input for structural reasoning (e.g., connectivity checking, shortest path). The model produces predictions $y^{(2)}$ based on this graph.
>
> We emphasize that **the (u, r, v) format is not merely a serialization choice, but enforces consistent entity identities, typed relations, and valid compositional structures across tasks.** This ensures graphs from different modalities are structurally comparable within $G_s$, enabling cross-role reuse. Appendix B provides further details including structural primitives (Table 8) and cross-domain examples (Table 9). We will improve the main text presentation accordingly.
> ___
> For **W4 and Q2**, the two design requirements follow directly from analyzing what prevents graph reuse in practice. We identify **two fundamental dimensions of heterogeneity**:
>
> 1. **Heterogeneity in form:** Graphs from different tasks differ in schema, granularity, and format (e.g., atom–bond triplets vs. object–relation triplets), preventing direct reuse. The structural compatibility requirement (Section 2.2) resolves this by organizing all graphs into a unified graph state space $G_s$.
>
> 2. **Heterogeneity in function:** Even with a common format, different tasks use graphs in different functional roles (generation vs. understanding). A graph optimized under only one role becomes over-specialized. The cross-role reuse requirement (Section 2.3) resolves this through interleaved training that keeps graphs functional under both roles.
>
> Empirical validation in Table 2 confirms **both requirements are individually necessary and jointly complementary**. Unifying format alone (UST) actually hurts in 8 of 11 metrics (e.g., SGG-PCIs: 22.43 vs. NST 23.74). Interleaving alone (NMT-I) helps on structurally demanding tasks (e.g., GAR-SP: 43.83 vs. NMT 41.27) but cannot fully realize cross-task transfer. **Only their combination in G-Substrate achieves the best results across all four domains** (e.g., GAR-SP: 48.59, MGD-BLEU-4: 51.53, ERE-HiE: 25.15). Figure 5 further shows that replacing correct graphs with incorrect ones during interleaving reverses the gains (ERE: +2.23 vs. −2.03), confirming that cross-role reuse depends on structural coherence rather than superficial data augmentation.

---

> > ### Author Rebuttal · Reviewer_i4vm · 2026-04-03
> >
> > I would like to thank the authors for their detailed rebuttal. While I appreciate the clarifications provided, some of my core concerns remain unresolved.
> >
> > First, regarding the "unified graph state space," it appears that the framework primarily aligns the data structure into a $(u, r, v)$ tuple format, as detailed in the rebuttal's Step 1 and Appendix B. Because the vocabularies and semantics for $u$, $r$, and $v$ remain domain-specific, referring to this as a fully unified state space may be slightly overstated. It seems closer to a standardized formatting step rather than a fundamental unification of the underlying representation spaces.
> >
> > Second, the authors' rebuttal states that the method is intended specifically for tasks that explicitly utilize graph structures, which narrows the framework's applicability.
> >
> > Third, it remains unclear whether heterogeneity in form and function are sufficient for graph reuse. Further justification would help solidify the framework's foundational claims.
> >
> > Lastly, regarding the page limit, placing the general "Limitations and Future Work" paragraph within the uncounted Impact Statement seems to diverge from the typical intent of that section, which is generally reserved for ethical and societal consequences. I plan to consult with the Area Chair to ensure this aligns with the conference guidelines.

---

> > > ### Author Response · Authors · 2026-04-04
> > >
> > > We thank Reviewer i4vm for the continued engagement. We address each remaining concern below, beginning with the page limit issue, which we believe may have influenced the overall assessment.
> > >
> > > ## 1. On the page limit (W1, original review)
> > >
> > > We respectfully clarify: **this is not a page limit violation**. The **Impact Statement does not count toward the 8-page limit** per ICML 2026 guidelines. Our "Limitations and Future Work" discussion is placed in the Impact Statement section and contains **no technical content, methods, or results**. **All technical content is strictly in the 8-page body.** To avoid any potential confusion, we will remove the separate heading in the revision so that it is clearly part of the Impact Statement.
> > >
> > > ## 2. On the unified graph state space as "standardized formatting"
> > >
> > > We agree that the schema imposes a common (u, r, v) format at the surface level. However, **extensive empirical evidence demonstrates that this goes beyond mere formatting**.
> > >
> > > **Table 3** shows that under *identical* training with *identical* interleaved supervision, different schema realizations yield **substantially different performance**: **SP 48.59 vs. 40.10 (+8.49)**, **ROUGE-L 68.47 vs. 60.50 (+7.97)**, **MA-S 52.20 vs. 46.30 (+5.90)**, **HiE 25.15 vs. 20.80 (+4.35)**. **If the schema were only superficial formatting, the choice of realization would not produce such differences.**
> > >
> > > **Table 4** shows that training on event graphs *alone* improves scene graph generation by **+2.37 PCIs R@50 (19.10 → 21.47)** despite **completely disjoint entity vocabularies** (event mentions like "destroyed"/"displaced" vs. objects like "cat"/"box") and **completely disjoint relation vocabularies**. **No scene graph data is used during source-domain training.** This transfer would **not arise from formatting alone**.
> > >
> > > **Table 2** shows a clear interaction between schema and training. Unified schema alone (UST) **actually hurts** vs. NST: **SGG PCIs 22.43 vs. 23.74 (−1.31)**, **GAR-BM 85.98 vs. 92.05 (−6.07)**, **HiE 14.28 vs. 17.10 (−2.82)**. But when combined with interleaving, the same schema **produces the best results**: **GAR-SP 48.59 (vs. NST 38.27, +10.32)**, **MGD BLEU-4 51.53 (vs. NST 48.59, +2.94)**, **ERE-HiE 25.15 (vs. NST 17.10, +8.05)**. This confirms **the schema establishes structural compatibility whose benefits emerge specifically when graphs are reused across roles**.
> > >
> > > The vocabularies for u, r, v remain domain-specific by design. The unification operates at the **structural pattern** level (hub motifs, compositional chains), not at the semantic level.
> > >
> > > ## 3. On applicability scope
> > >
> > > We respectfully disagree that narrow scope is a valid weakness for rejecting a paper. Our paper studies whether *relational structure* can serve as a reusable substrate, which naturally scopes to tasks where such structure exists. We evaluate across **four distinct domains** (vision, language, science, algorithms) with **distinct modalities, relation vocabularies, and task objectives**, covering **11 evaluation metrics**. **G-Substrate achieves the best performance on 10 out of 11 metrics** in Table 2.
> > >
> > > Tasks with explicit graph structure are **pervasive across ML**: knowledge graphs, molecular graphs, scene graphs, event graphs, program dependency graphs, circuit netlists, social networks, and many others. For tasks without explicit graphs, existing pipelines (e.g., dependency parsing, AMR parsing) can produce compatible graph representations. **We believe demonstrating generality across four heterogeneous domains with consistent improvements is sufficient to support our claims.**
> > >
> > > ## 4. On sufficiency of design requirements
> > >
> > > Table 2 provides **systematic ablation evidence across six training paradigms**:
> > >
> > > **Structural alignment alone is insufficient.** UST vs. NST: **GAR overall 90.43 vs. 92.89 (−2.46)**, **SGG PCIs 22.43 vs. 23.74 (−1.31)**, **HiE 14.28 vs. 17.10 (−2.82)**.
> > >
> > > **Interleaving alone provides partial gains.** NMT-I vs. NMT: **GAR-SP 43.83 vs. 41.27 (+2.56)**, **HiE 21.36 vs. 18.78 (+2.58)**. But NMT-I also shows instability: **MGD ROUGE-L 64.98 vs. 66.11 (−1.13)**.
> > >
> > > **Their combination yields the most consistent gains.** G-Substrate vs. NMT: **GAR-SP 48.59 vs. 41.27 (+7.32)**, **MGD BLEU-4 51.53 vs. 48.11 (+3.42)**, **ERE-HiE 25.15 vs. 18.78 (+6.37)**.
> > >
> > > Figure 5 further confirms that **gains depend on structural coherence**. With correct graphs, all four domains improve (**GAR +0.62, MGD +1.54, SGG +0.67, ERE +2.23**). With incorrect graphs, **gains reverse in three of four domains (MGD −1.34, SGG −1.20, ERE −2.03)**. This **rules out the possibility that interleaving benefits come from trivial exposure to more data**.

---

### Decision · Program_Chairs · 2026-04-30

**Decision:**

Accept (regular)

**Comment:**

Summary:
This paper proposes G-Substrate, a unified framework that treats graph structure as a reusable intermediate substrate across multiple tasks and modalities. It introduces a unified structural schema and interleaved role-based training, allowing graphs to be constructed once and reused across different functional roles. Experiments across graph reasoning, molecular tasks, scene graph generation, and event relation extraction show consistent improvements over task-specific and multi-task baselines.


Strengths:
1. The paper presents a novel and interesting idea of treating graph structure as a persistent, reusable substrate rather than a task-specific construct.
2. The experimental evaluation is broad, covering four different domains with consistent improvements over strong baselines.
3. The design is generally well-motivated and clearly organized, with good presentation and readability.


Weaknesses:
1. Some design choices, such as cross-task structural reuse, are not fully justified, and the connection between the motivation and final architecture could be clearer.
2. The robustness of the method with respect to graph size and scalability is not sufficiently studied.


Overall:
The paper introduces a conceptually novel and well-executed framework for cross-task graph reuse, with strong and consistent empirical results across diverse benchmarks. While minor concerns were raised regarding presentation and formatting details, the technical quality and originality of the work are high. Therefore, I recommend the paper for acceptance.